



# Spatially dependent Intensity-Duration-Frequency curves to support the design of civil infrastructure systems

**Phuong Dong Le[1,2], Michael Leonard[1], Seth Westra[1]**

[1]*School of Civil, Environmental and Mining Engineering, University of Adelaide, Adelaide, South Australia, Australia*

[2]*Thuyloi University, Hanoi, Vietnam*

*Email: phuongdong.le@adelaide.edu.au*

**Keywords: areal reduction factor, asymptotic independence, conditional probability, duration dependence, extreme rainfall, flood probability, inverted max-stable process, joint probability, spatially dependent Intensity-Duration-Frequency,**

**Abstract**

Conventional flood risk methods typically focus on estimation at a single location, which is inadequate for civil infrastructure systems such as road or railway infrastructure. This is because rainfall extremes are spatially dependent, so that to understand overall system risk it is necessary to assess the interconnected elements of the system jointly. For example, when designing evacuation routes it is necessary to understand the risk of one part of the system failing given that another region is flooded or exceeds the level at which evacuation becomes necessary. Similarly, failure of any single part of a road section (e.g., a flooded river crossing) may lead to the wider system's failure (i.e. the entire road becomes inoperable). This study demonstrates a spatially dependent Intensity-Duration-Frequency curve framework that can be used to estimate flood risk across multiple catchments, accounting for dependence both in space and across different critical storm durations. The framework is demonstrated via a case study of a highway upgrade, comprising five bridge crossings where the upstream contributing catchments each have different times of concentration. The results show that conditional and unconditional design flows can differ by a factor of two, highlighting the importance of taking an integrated approach. There is also a reduction in the failure probability of the overall system compared with the case of no spatial dependence between storms. The results demonstrate the



potential uses of spatially dependent Intensity-Duration-Frequency curves and suggest the need for
more conservative design estimates to take into account conditional risks.



## 1. Introduction

Methods for quantifying flood risk of civil infrastructure systems such as road and rail networks
require considerably more information compared to traditional methods that focus on flood risk at a
point. For example, the design of evacuation routes requires the quantification of the risk that one part
of the system will fail at the same time that another region is flooded or exceeds the level at which
evacuation becomes necessary. Similarly, a railway route may become impassable if any of a number
of bridges are submerged, such that the 'failure probability' of that route becomes some aggregation
of the failure probabilities of each individual section. Successful estimation of flood risk in these
systems therefore requires recognition both of the networked nature of the civil infrastructure system
across a spatial domain, as well as the spatial and temporal structure of flood-producing mechanisms
(e.g. storms and extreme rainfall) that can lead to system failure (e.g., Leonard et al. (2014),
Seneviratne et al. (2012), Zscheischler et al. (2018)).
One way to estimate such flood probabilities is to directly use information contained in historical
streamflow data. For example, annual maximum streamflow at two locations might be assumed to
follow a bivariate generalized extreme value distribution (Favre et al., 2004; Wang, 2001; Wang et al.,
2009), which can then be used to estimate both conditional probabilities (e.g. the probability that one
river is flooded given that the other river level exceeds a specified threshold) and joint probabilities
(e.g. the probability that one or both rivers are flooded). However, continuous streamflow data are
often not available at the locations most relevant to the civil infrastructure system in question, or the
catchment conditions have changed to a degree that reflects historical streamflow records as
unrepresentative of likely future risk. Thus, direct application of streamflow data for flood risk
quantification in civil infrastructure systems does not represent a viable approach for the majority of
situations.
To deal with these difficulties, two alternative rainfall-based approaches are commonly used. The first
uses continuous rainfall data (either historical or generated) to compute continuous streamflow data
using a rainfall-runoff model (Boughton and Droop, 2003; Cameron et al., 1999; He et al., 2011;
Hegnauer et al., 2014; Pathiraja et al., 2012), with flood risk then estimated based on the simulated



streamflow time series. This method is computationally intensive and given the challenge of
reproducing a wide variety of statistics across many scales, can have difficulties in modelling the
dependence of extremes. Most rainfall models operate at the daily timescale (Baxevani and
Lennartsson, 2015; Bennett et al., 2016b; Hegnauer et al., 2014; Kleiber et al., 2012; Rasmussen,
2013), whereas many catchments respond at subdaily timescales. The capacity of space-time rainfall
models to simulate the statistics of sub-daily rainfall remains a challenging research problem (Leonard
et al., 2008). One approach is to exploit the relative abundance of data at the daily scale, then apply a
downscaling model to reach subdaily scales (Gupta and Tarboton, 2016). Continuous simulation is
receiving ongoing attention and increasing application, yet there remain limitations when applying
these models in many practical contexts.
The second rainfall-based approach proceeds by conducting the probability calculations on rainfall, to
construct 'Intensity-Duration-Frequency' (IDF) curves, which are then translated to a runoff event of
equivalent probability via either empirical models such as the Rational method (Kuichling, 1889;
Mulvaney, 1851) to estimate peak flow rate, or via event-based rainfall-runoff models that are able to
simulate the full flood hydrograph (Boyd et al., 1996; Chow et al., 1988; Laurenson and Mein, 1997).
Currently IDF curves are estimated either at a point location, or are estimated over a spatial domain
by multiplication with an areal reduction factor (ARF) to convert point rainfall to spatially averaged
rainfall of an equivalent exceedance probability (Ball et al., 2016); this information then can be used
to estimate either peak flow or the flood hydrograph at any point location within a catchment.
However, such methods do not account for information on the spatial dependence of extreme
rainfall—whether for single storm duration across a region, or for the more complex case of different
durations across a region (Bernard, 1932; Koutsoyiannis et al., 1998). This prevents these approaches
from being applied to estimate conditional or joint flood risk at multiple points in a catchment or
across several catchments as would be required for a civil infrastructure system.
Although tailored multivariate approaches can be applied to estimate conditional and joint
probabilities of extreme rainfall for specific situations (e.g., Kao and Govindaraju (2008), Wang et al.
(2010), Zhang and Singh (2007)), the development of a unified methodology that integrates with



existing IDF-based flood estimation approaches remains elusive. This is particularly challenging
given that it is not only necessary to preserve dependence of rainfall across space, but also to account
for dependence across storm burst durations, as different parts of the system may be vulnerable to
different critical duration storm events. To this end, arguably the most promising recent research
direction has been the application of max-stable process theory that is able to represent storm-level
dependence (de Haan, 1984; Schlather, 2002). This has been applied on a spatial domain by Padoan et
al. (2010), who calculated conditional probabilities for a spatial domain located in United States.
However, to ensure that this general approach can be applied for practical flood estimation problems,
two further problems need to be overcome:
1.  The approach needs to not only account for spatial dependence for rainfall 'events' of a single

duration (e.g. the field of annual maximum daily rainfall data), but must also account for

dependence across multiple durations. This was addressed by Le et al. (2018b), who linked

the max-stable model of Brown and Resnick (1977) and Kabluchko et al. (2009) with the

duration-dependent model of Koutsoyiannis et al. (1998), in order to create a model that could

be used to reflect dependencies between nearby catchments of different sizes.

2.  Given that often the interest is in rare flood events, the model needs to capture appropriate
asymptotic properties of spatial dependence as the events become increasingly extreme.
Recent evidence is emerging that rainfall has an asymptotically independent characteristic (Le
et al., 2018a; Thibaud et al., 2013), which means that the level of the rainfall's dependence
reduces with an increasing return period (Wadsworth and Tawn, 2012). This implies that
inverted max-stable models, which are asymptotically independent, are likely to be preferable
as an approach for representing spatially dependent IDF information. An added benefit of
correctly representing asymptotic dependence is that information on areal reduction factors
can be obtained directly from the model, rather than estimating ARF information
independently from the computation of the IDF curves.
This study addresses both these issues by demonstrating the application of the inverted max-stable
process to estimate joint and conditional probabilities of flood-producing rainfall in the form of



spatially dependent IDF curves. This approach adapts the methods developed by (Le et al., 2018b) to
inverted max-stable models, and then uses the derived spatially-dependent IDF curves combined with
the extracted information on AFRs as the basis for transforming the rainfall into flood flows. The
approach is demonstrated on a highway system spanning 20 km with five separate bridge crossings,
and with the contributing catchment at each crossing having a different time of concentration.
The case study is designed to address two related questions: (i) "What flood flow needs to be used to
design a bridge that will fail only once on average every $M$ times (e.g., $M = 10$ for a 10-year event)
that a neighbouring catchment is flooded?"; and (ii) "What is the probability that the overall system
fails given that each bridge is designed to a specific exceedance probability event (e.g., the 1% annual
exceedance probability event)?" The method for resolving these questions represents a new paradigm
in which to estimate flood risk for engineering design, by focusing attention on the risk of the entire
system, rather than the risk of individual system elements in isolation.
In the remainder of the paper, Section 2 emphasises the need for spatially dependent IDF curves in
flood risk design, followed by Section 3 which outlines the case study and data used. Section 4
explains the methodology of the framework, including a method for analysing the spatial dependence
of extreme rainfall across different durations. It also includes an algorithm with which to use that
information in estimating the conditional and joint probabilities of floods. The results, and a
discussion on the behaviour of flood due to the spatial and duration dependence of rainfall extremes,
are provided in Section 5. Conclusions and recommendations follow in Section 6.
**2. The need for spatially dependent IDF curves in flood risk estimation**
The main limitation of conventional methods of flood risk estimation is that they isolate bursts of
rainfall and break the dependence structure of extreme rainfall. Figure 1 demonstrates a traditional
process of estimating at-site extreme rainfall for two locations (gauge 1, gauge 2) and three durations
(1, 3, and 5 hr) (Stedinger et al., 1993). The process first involves extracting the extreme burst of
rainfall for each site, duration and year from the continuous rainfall data, and then fitting a probability
distribution (such as the Generalised Extreme Value (GEV) distribution) to the extracted data. Figure
1 demonstrates that, through the process of converting the continuous rainfall data to a series of



discrete rainfall 'bursts', this process breaks both the dependence with respect to duration and space.
Firstly, the duration dependence is broken by extracting each duration separately, whereas for the
hypothetical storm in Fig. 1 it is clear that the annual maxima from some of the extreme bursts come
from the same storm. Secondly, the spatial dependence is broken because each site is analysed
independently. Again, for the hypothetical storm of Fig. 1 it can be seen that the 5 hr storm has
occurred at the same time across the two catchments, and this information is lost in the subsequent
probability distribution curves. Lastly, there is cross-dependence in space and duration. For example,
the 1 hr extreme from gauge 2 occurs at the same time as the 5 hr extreme from gauge 1. This may be
relevant if there are two catchments with times of concentration matching 1 hr and 5 hr respectively,
where catchments are neighbouring or nested.

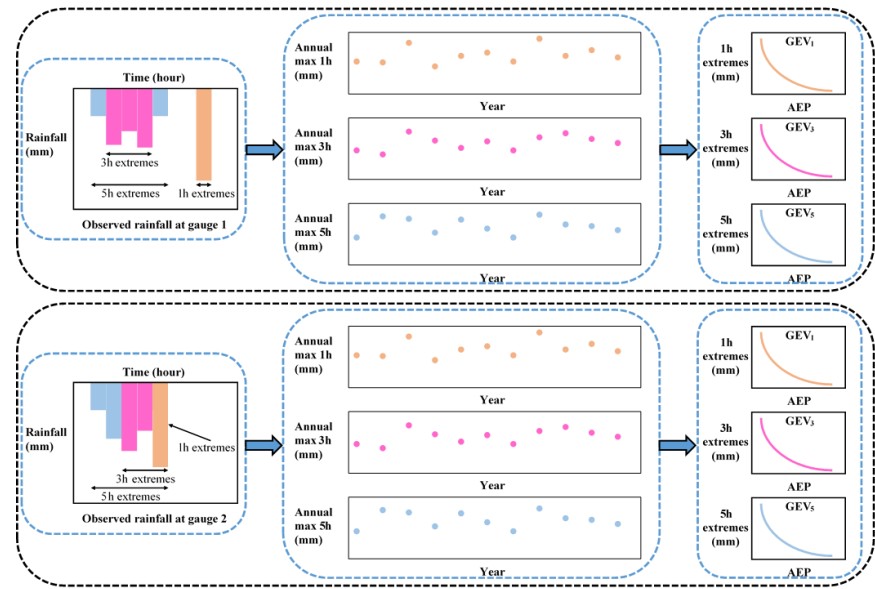


**Figure 1.** Illustration of process to estimate rainfall extremes for each individual location in conventional flood risk

approach, the upper panel is for gauge 1 and the lower panel is for gauge 2.

Having obtained the IDF curves for individual locations in Fig. 1, the next step is commonly to
convert this to spatial IDF maps by interpolating results between gauged locations. Figure 2 shows
hypothetical IDF curves from individual sites, with a separate spatial contour map usually provided
for each storm burst duration. In a conventional application the respective maps are used to estimate



the magnitude of extreme rainfall over catchments for a specified time of concentration. The IDF
curves are combined with an areal reduction factor (ARF) to determine the volume of rainfall over a
region (since rainfall is not simultaneously extreme at all locations over the region). However,
because the spatial dependence was broken in the analysis of IDF curves, the ARF come from a
separate analysis and are an attempt to correct for the broken spatial relationship within a catchment
(Bennett et al., 2016a). Lastly, the rainfall volume over the catchment is combined with a temporal
pattern and input to a runoff model to simulate flood-flow at a catchment's outlet. Where catchment
flows can be considered independently this process has been acceptable for conventional design, but
because this process does not account for dependence across durations and across a region, it is not
possible to address problems that span multiple catchments, as with civil infrastructure systems.

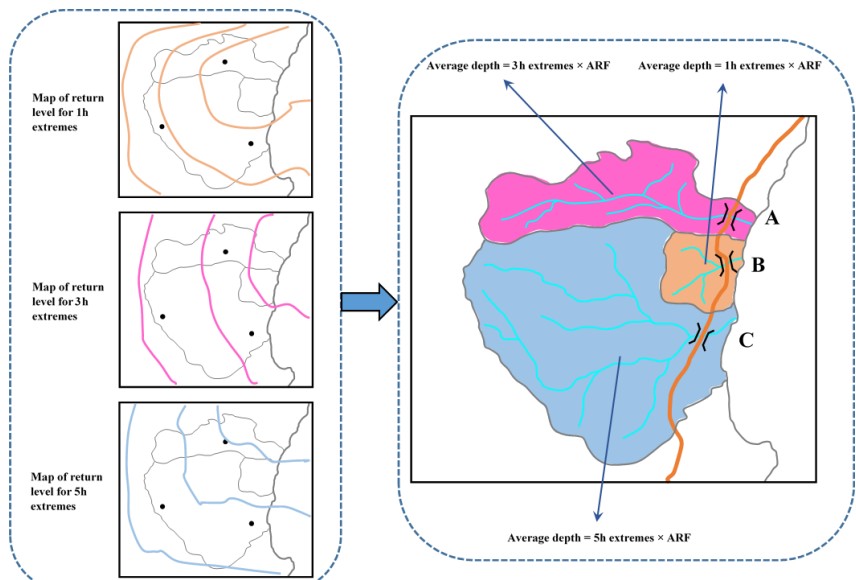


**Figure 2.** Illustration of map of return level and how to use it in estimating flood flow in conventional flood risk estimates
approach.
The process in Fig. 1 breaks out the dependence of the observed rainfall, which makes the
conventional approach unable to analyse the dependence of flooding at two or more separate
locations. Instead, this paper advocates for spatially dependent IDF curves which are developed by





retaining the dependence of observed rainfall in the estimation of extremal rainfall. By applying
spatially dependent IDF curves to a rainfall-runoff model, the dependence of flooding between
separate locations can be achieved.
**3. Case study and data**
The region chosen for the case study is in the mid north coast region of New South Wales, Australia.
This region has been the focus of a highway upgrade project and has an annual average daily traffic
volume on the order of 15,000 vehicles along the existing highway. The upgrade traverses a series of
coastal foothills and floodplains for a total length of approximately 20 km. The project's major river
crossings consist of extensive floodplains with some marsh areas.
The case study has five main catchments that are numbered in sequence in Fig. 3: (1) Bellinger, (2)
Kalang River, (3) Deep Creek, (4) Nambucca and (5) Warrell Creek. The area and time of
concentration of these catchments is summarised in Table 1, with the latter estimated using the ratio
of the flow path length and average flow velocity (SKM, 2011). The Deep Creek catchment has a time
of concentration of 8.3 hr, while the other four catchments have much longer times of concentration,
ranging from 27.8 to 38.9 hr. These require the estimates of spatial dependence across different
durations of rainfall extremes. Although the spatial dependence across rainfall durations would be
expected to be lower than across a single duration, since short- and long-rain events are often driven
by different meteorological mechanisms (Zheng et al., 2015), it is nonetheless likely that some level
of spatial dependence would exist and need to be integrated into the risk calculations. This is
particularly of relevance given extremal rainfall in this region is strongly associated with 'east coast
low' systems off the eastern coastline, whereby extreme hourly rainfall bursts are often embedded in
heavy multi-day rainfall events.



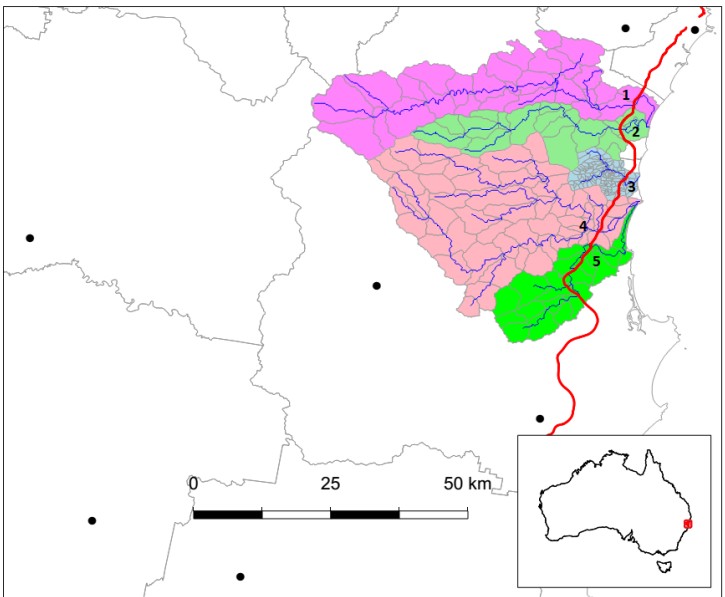


**Figure 3.** Map of the case study in New South Wales, Australia. The black dots indicate the rainfall gauges, the red line indicates the Pacific Highway upgrade project, and the blue lines indicate the main river network. The numbers from one to five indicate the locations of the main river crossings.

**Table 1.** Summary of properties for catchments in the case study.

| No. | Catchment | Area (ha) | Raw time of concentration (hour) |
|-----|-----------|-----------|----------------------------------|
| 1 | Bellinger | 77150 | 37 |
| 2 | Kalang River | 34140 | 33 |
| 3 | Deep Creek | 9180 | 8 |
| 4 | Nambucca (upper) | 102015 | 38 |
| 5 | Warrell Creek | 29440 | 27 |

The black circles in Fig. 3 represent the sub-daily rain stations used for this study. There were 7 sub-daily stations selected, with 35 years of record in common for the whole region. The data was available at a 5 minute interval and aggregated to longer durations. For convenience in comparing the times of concentration between the catchments, this study assumes a time of concentration of 9 hr for the Deep Creek catchment, while identical times of concentration of 36 hr are assumed for the other four catchments.




## 4. Methodology

This section provides the method used to estimate the conditional and joint probabilities of flood for civil infrastructure systems based on rainfall extremes, which is explained according to the steps shown in Fig. 4. First, the generalized Pareto distribution (GPD) is used as marginal distribution to fit to observed rainfall for all duration at each locations (Section 4.1). After that, an inverted max-stable process is introduced and then fitted to rainfall extremes of identical or different durations (Sections 4.2 & 4.3). The conditional and joint probabilities of rainfall are then estimated in Section 4.4, which is followed by the simulation to calculate areal reduction factor (ARF) in Section 4.5. An event-based rainfall-runoff is employed in Section 4.6 to transform conditional rainfall to conditional flows. With an assumption of there is a one-to-one correspondence between rainfall intensity and flow rate, the joint flood probability for the case study is equal to the joint probability of rainfall. An analysis for the independent model (the case of complete independence) is also implemented for comparison.

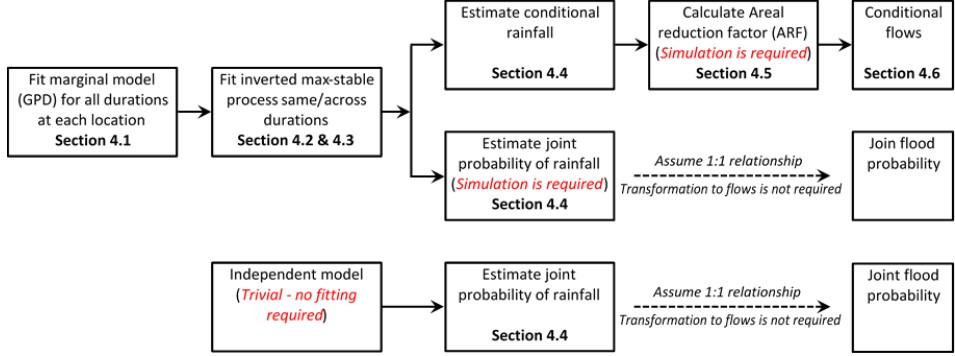

**Figure 4.** The flow chart for the overall methodology.

### 4.1. Marginal model for rainfall

This study defines extremes as those greater than some threshold $u$. For large $u$, the distribution of $Y$ conditional on $Y > u$ may be approximated by the generalized Pareto distribution (GPD) (Davison and Smith, 1990; Pickands, 1975; Thibaud et al., 2013):

$$G(y) = 1 - \left\{ 1 + \frac{\xi(y - u)}{\sigma_u} \right\}^{-1/\xi}, \quad y > u, \tag{1}$$





defined on $\{y: 1 + \xi(y - u)/\sigma_u > 0\}$ where $\sigma_u > 0$ and $-\infty < \xi < +\infty$ are scale and shape
parameters, respectively. The probability that a level $y$ is exceeded is then $\Phi_u\{1 - G(y)\}$, where
$\Phi_u = \Pr(Y > u)$.
The selection of the appropriate threshold $u$ involves a trade-off between bias and variance. A
threshold that is too low leads to bias because the GPD approximation is poor. A threshold too high
leads to high variance because of a small number of excesses. Two diagnostic tests are used to
determine the appropriate threshold $u$: the mean residual life plot and the parameter estimate plot
(Coles, 2001; Davison and Smith, 1990). These methods use the stability property of a GPD, so that if
a GPD is valid for all excesses above $u$, then excesses of a threshold greater than $u$ should also follow
a GPD. Detailed guidance of these methods can be found in Coles (2001).
*4.2. Dependence model for spatial rainfall*
Consider rainfall as a stationary stochastic process $Z_i$ associated with a location $x_i$ in a region of
interest. Models for spatial extremes often use the convention of transforming marginal values to a
unit Fréchet distribution. An important property of dependence in the extremes is whether or not two
variables are likely/unlikely to co-occur as the extremes become rarer, as this can significantly
influence the estimate of frequency for flood events of large magnitude. This is referred to as
asymptotic dependence/independence, respectively. For the case of asymptotic independence, the
dependence structure becomes weaker as the extremal threshold increases, which is formally defined
as $\lim_{z \to \infty} P\{Z_1 > z | Z_2 > z\} = 0$ for all $x_1 \neq x_2$. The spatial extent of a rainfall event with
asymptotically independent extremes will diminish as its rarity increases.
An example of an asymptotically independent model is the inverted max-stable process (Wadsworth
and Tawn, 2012). This study uses the Brown-Resnick form of equations from the family of an
inverted max-stable process, and has been widely studied elsewhere (Asadi et al., 2015; Huser and
Davison, 2013; Kabluchko et al., 2009; Oesting et al., 2017).
*4.3. Fitting the dependence model*




One simple way to calibrate dependence models is to fit them to data by matching a suitable statistic.
The dependence structure of the inverted max-stable process is represented by the pairwise residual
tail dependence coefficient (Ledford and Tawn, 1996).
For a generic continuous process $Z_i$ associated with a specific location $x_i$ the empirical pairwise
residual tail dependence coefficient $\eta$ for each pair of locations $(x_1, x_2)$ is

$$\eta(x_1, x_2) = \lim_{y \to \infty} \frac{\log P\{Z_2 > z\}}{\log P\{Z_1 > z, Z_2 > z\}}. \tag{2}$$

The value of $\eta \in (0,1]$ indicates the level of extremal dependence between $Z_1$ and $Z_2$ (Coles et al.,
1999), with lower values indicating lower dependence. An example of how to calculate the residual
tail dependence coefficient is provided in Appendix A for a sample dataset.
To estimate the dependence structure of an inverted max-stable model, the theoretical residual tail
dependence coefficient function is usually fitted to its empirical counterpart. Here the residual tail
dependence coefficient function is assumed to only depend on the Euclidean distance between two
locations $h = \|x_1 - x_2\|$. The theoretical residual tail dependence coefficient function for the inverted
Brown-Resnick model is given as:

$$\eta(h) = \frac{1}{2\Phi\left\{\sqrt{\frac{\gamma(h)}{2}}\right\}}, \tag{3}$$

where $\Phi$ is the standard normal cumulative distribution function, $h$ is the distance between two
locations, and $\gamma(h)$ belongs to the class of variograms $\gamma(h) = \|h\|^{\beta}/q$ for $q > 0$ and $\beta \in (0,2)$. The
models are then fitted to the empirical residual tail dependence coefficients by modifying parameters
$q$ and $\beta$ until the sum of squared errors is minimized.
In the case that extreme rainfall at locations $x_1$ and $x_2$ are of identical duration (i.e. both 36 hr), then
the inverted max-stable process is fitted to the observations by minimizing the sum of the squared
errors of the residual tail dependence coefficients. This information can be directly applied to the case
where two catchments have a similar time of concentration owing to their similar shape and size.
However, there are many instances when two catchments of interest will have differing times of



concentration; in particular, when the extreme rainfall at location $x_1$ and $x_2$ are of different durations
(e.g., 36 hr and 9 hr), the dependence is less than the case of 36 hr and 36 hr. This observation is
evident when considering the special case of a single location, i.e. the same point is considered twice,
at a distance of $h = 0$. For the case where the duration is the same where, the rainfall values are
identical and have perfect dependence, but when the duration of extremes are different the values are
not identical and the dependence is less. Therefore, an adjustment needs to be made to ensure that the
theoretical pairwise residual tail dependence coefficient function suitably represents the observed
pairwise residual tail dependence coefficients for the case of extreme rainfalls of different durations.
Following Le et al. (2018b), an adjusted approach is used by adding a nugget to the variograms as:

$$\gamma_{ad.}(h) = h^\beta/q + c(D - d)/d, \tag{4}$$

where $h$, $\beta$, and $q$ are the same as those in Eq. (3); $d$ is the duration (in hours); $0 < d \leq D$, where $D$ is
the maximum duration of interest (e.g. $D = 36$ hr for the case study described in this paper); and $c$ is
a parameters to adjust dependence according to duration. This adjustment is intended to condition the
behaviour of shorter duration extremes on a $D$-hour extreme of a specified magnitude. It is
constructed to reflect the fact that when compared to a $D$-hour extreme, a shorter duration results in
less extremal dependence. Cases involving conditioning of longer periods on shorter periods (such as
a 36 hr extreme given a 9 hr extreme has occurred) would require a different relationship.
To fit the inverted max-stable process for all pairs of durations at locations $x_1$ and $x_2$ (i.e. 36 hr and
12 hr, 36 hr and 9 hr, 36 hr and 6 hr, 36 hr and 2 hr, 36 hr and 1 hr), the theoretical pairwise residual
tail dependence coefficient function in Eq. (3) is used with the adjusted variogram from Eq. (4) where
the parameters $\beta$ and $q$ are first obtained from the fitted results of the case of identical 36 hr durations
at location $x_1$ and $x_2$. The parameter $c$ is obtained by a least square fit of the residual tail dependence
coefficient across all durations.
***4.4. Estimate of conditional and joint probabilities of rainfall extremes***



This section introduces general concepts for evaluating a conditional probability and a joint
probability for a bivariate case. A detailed method is then presented for estimating the conditional
probability and the joint probability for the realistic case of rainfall extremes.
Figure 5 illustrates a bivariate case for two locations $x_1$ and $x_2$ as a scatterplot of events at two
locations. The extremes are delineated for each location according to a specified threshold (e.g.
$u = 0.98$ percentile) and to distinguish them, colour coding and different symbols have been used. The
four regions have been labelled for ease of reference: (A) only $Z_2$ extreme events but not $Z_1$, (B) both
$Z_1$ and $Z_2$   extreme, (C) only $Z_1$ extreme events but not $Z_2$, and (D) non-extreme events.

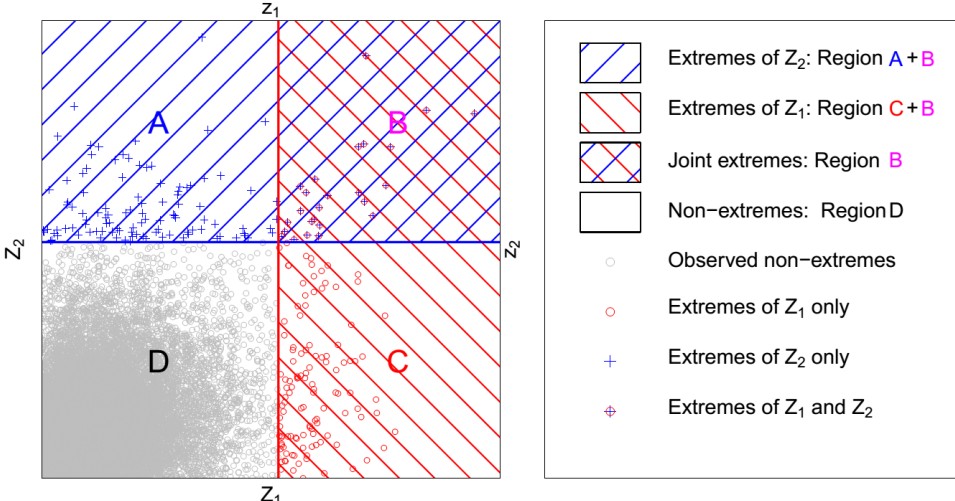


**Figure 5.** Illustration of general concept of probabilities for a bivariate case. $Z_1$ and $z_1$ indicate stochastic process Z and a
threshold at location $x_1$; $Z_2$ and $z_2$ indicate stochastic process Z and a threshold at location $x_2$.

To explain how the joint and conditional probabilities are calculated, their definitions are provided in
Table 2 with reference to the regions of Fig. 5. Rather than consider the specific case of a theoretical
model of extremal rain (e.g. inverted max stable), Table 2 presents these concepts more simply using
only two variables and with generic probability estimates. Equations for both dependence and
independence are provided in Table 2.
**Table 2.** Definition of joint and conditional probabilities and how to calculate them for the case of bivariate independent and
dependent variables.





| Case | Definition | Calculation |
|---|---|---|
| 1. Conditional prob. dependent | $P\{Z_2 > z_2 \mid Z_1 > z_1\}$ | $= P(B)/\{P(B) + P(C)\}$ |
| 2. Conditional prob. independent | $P\{Z_2 > z_2 \mid Z_1 > z_1\} = P\{Z_2 > z_2\}$ | $= P(A) + P(B)$ |
| 3. Joint prob. dependent | $P\{Z_1 > z_1, Z_2 > z_2\}$ | $= P(B)$ |
| 4. Joint prob. independent | $P\{Z_1 > z_1, Z_2 > z_2\} = P\{Z_1 > z_1\} \times P\{Z_2 > z_2\}$ | $= \{P(B) + P(C)\}\{P(A) + P(B)\}$ |

**Case 1**: Conditional probability can be defined as the joint probability divided by the marginal
probability $P\{Z_2 > z_2 \mid Z_1 > z_1\} = P\{Z_1 > z_1, Z_2 > z_2\}/P\{Z_1 > z_1\}$. For the dependent case, the
relationship is $P(B)/\{P(B) + P(C)\}$. Using these concepts, equations for the conditional probability
of the inverted max-stable process have been derived in literature and are summarised in Appendix B.
The detailed formulae are of the same nature as those in Table 2, and are used in this study to estimate
conditional maps for return periods once the model has been fitted to all durations.
**Case 2:** Using the definition of $P\{Z_2 > z_2 \mid Z_1 > z_1\} = P\{Z_1 > z_1, Z_2 > z_2\}/P\{Z_1 > z_1\}$ for the
independent case results in the exceedance probability for $Z_2$, which is $P(A) + P(B)$ (since intuitively
$Z_1$ has no effect on exceedances of $Z_2$).
**Case 3:** For the case of dependent variables the joint exceedance is defined by $P(B)$. For the case of
only two locations, the probability that there is at least one location that has an extreme event
exceeding a given threshold is calculated as $P\{Z_1 > z_1 \text{ or } Z_2 > z_2\} = P\{Z_1 > z_1\} + P\{Z_2 > z_2\} -$
$P\{Z_1 > z_1, Z_2 > z_2\}$. Here, $P\{Z_1 > z_1, Z_2 > z_2\}$ can be easily obtained from the bivariate CDF for
inverted max-stable process in Eq. (B.1). However, for the case of multiple locations (five different
locations for this paper), it is difficult to derive the formula for this probability because there are
dependences between extreme events at all locations. So this probability is empirically calculated
from a large number of simulations of the dependent model (see the description of the simulation
procedure for an inverted max-stable process in Section 4.5). It is also noted that the case study
contains five catchments, which have approximate times of concentration of either 36 hr or 9 hrs.
**Case 4:** Joint probability for independent variables is broken down as the product of the marginals.
The exceedance probability for $Z_1$ is $P(B) + P(C)$ and the exceedance probability for $Z_2$ is $P(A) +$
$P(B)$, and by definition their independent product will result in the joint probability. In order to





compare with a situation of no spatial dependence of rainfall extremes, the probability that there is at
least one location that has an extreme event exceeding a given threshold for the case that all of events
are independent can be calculated based on the addition rule for the union of probabilities, as:
$$P(Z_1 > z_1 \text{ or } \dots \text{ or } Z_N > z_N) = \sum_{i=1}^{N} P(Z_i > z_i) - \sum_{i<j} P(Z_i > z_i, Z_j > z_j) + \cdots$$

$$+(-1)^{N-1}P(Z_1 > z_1, \dots, Z_N > z_N), \qquad (5)$$

where $N$ is the number of locations, and $P(Z_1 > z_1, \dots, Z_N > z_N) = P(Z_1 > z_1) \dots P(Z_N > z_N)$,
because all of the events are independent.
*4.5. Areal reduction factor estimation and simulation procedure for spatial rainfall*
Before being transformed to flood flow through an event-based model, the rainfall extremal estimates
need to be converted to the average spatial rainfall using an areal reduction factor (ARF) (Ball et al.,
2016). ARFs can be estimated from observed rainfall data, but it is difficult to extrapolate ARFs for
long return periods from observations with just 35 years of record for this study. To deal with this
difficulty and to analyse the asymptotic behaviour of ARFs, Le et al. (2018a) proposed a framework
to simulate ARFs for long return periods by using an inverted max-stable process, which is applied
here for durations of 36 and 9 hrs.
The simulation procedure for spatial rainfall is implemented in two steps. In the first step, the Brown-
Resnick process with unit Fréchet margins is simulated using the algorithm of Dombry et al. (2016)
over a spatial domain (whether specific locations of interest or grid points), and then the inverted
Brown-Resnick process with unit Fréchet margins is obtained through Eq. (4) and Eq. (5) in Le et al.
(2018a). In the second step, the spatial rainfall processes are obtained by transforming the simulation
of the inverted Brown-Resnick process in step 1 from unit Fréchet margins to the rainfall scaled
margins using the GP distribution in Eq. (1) for rainfall magnitude above the threshold, and the
empirical distribution for rainfall magnitude below the threshold. An advantage of this approach is
that it can reflect the proportion of dry days in the empirical distribution by making the simulated
rainfall contain zero values (Thibaud et al., 2013). Another advantage is that this approach guarantees



363 that the marginal distributions of simulated rainfall below the threshold matches the observed

364 marginal distributions. There may be a drawback of this approach by forcing the simulated rainfall to

365 have the same extremal dependence structure for both parts below and above the threshold, which

366 may not be true for non-extreme rainfall. However, the dependence structure of non-extreme rainfall

367 contributes insignificantly to extreme events (Thibaud et al., 2013) and is unlikely to affect the results.

368 For calculating ARFs, the simulation is implemented separately for spatial rainfall of 36 and 9 hrs

369 duration. After the simulated spatial rainfall for 36 and 9 hrs are respectively obtained, ARFs are

370 calculated for each duration and different return periods, which can be found in the supplementary

371 material (Fig. S1 and S2). When the interest is in the joint probability of rainfall extremes of different

372 durations (see Case 3 in Section 4.4), the simulation of spatial rainfall should be implemented across

373 multiple durations. In this case, each term of the covariance matrix is calculated from the dependence

374 structure of the corresponding pair of locations.

375 *4.6. Transforming rainfall extremes to flood flow*

376 To estimate flood flow from rainfall extremes, the Watershed Bounded Network Model (WBNM)

377 (Boyd et al., 1996), is employed in this study. WBNM calculates flood runoff from rainfall

378 hyetographs. It divides the catchment into subcatchments, allowing hydrographs to be calculated at

379 various points within the catchment, and allowing the spatial variability of rainfall and rainfall losses

380 to be modelled. It separates overland flow routing from channel routing, allowing changes to either or

381 both of these processes, for example in urbanised catchments. The rainfall extremes are estimated at

382 the centroid of the catchment, and are converted to average spatial rainfall using the simulated ARFs

383 described in Section 4.5 before estimation of the rainfall hyetographs.

384 Hydrological models for the case study area were developed and calibrated by engineering consultants

385 (WMAWater, 2011). As an example, Fig. 6 provides details of the hydrological models for the

386 Bellinger catchment and Kalang River catchment in the North. The plots for details of the

387 hydrological models for the Nambucca basin in the South and the Deep Creek catchment in the East

388 can be found in the supplementary material (Fig. S3 and S4).





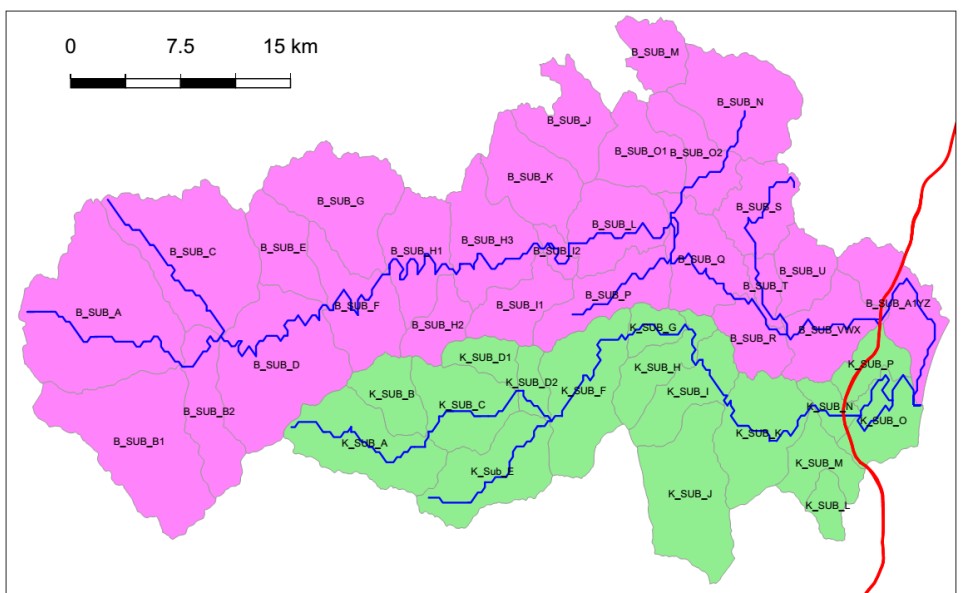

**Figure 6.** Hydrological model layout for Bellinger catchment and Kalang River catchment. The blue lines are the river network, and the red line is the Pacific Highway upgrade project.

## 5. Results and discussion

### 5.1. Evaluation of model for space-duration rainfall process

A GPD with an appropriate threshold was fitted to the observed rainfall data for 36 hr and 9 hr durations, and the Brown-Resnick inverted max-stable process model was calibrated to determine the spatial dependence.

Analysis of the rainfall records led to the selection of a threshold of 0.98 for all records as reasonable across the spatial domain and the GPD was fitted to data above the selected threshold. Figure 7 shows QQ plots of the marginal estimates for a representative station for two durations 36 and 9 hr. Overall the quality of fitted distributions is good and plots for all other stations can be found in the supplementary material (Fig. S5).





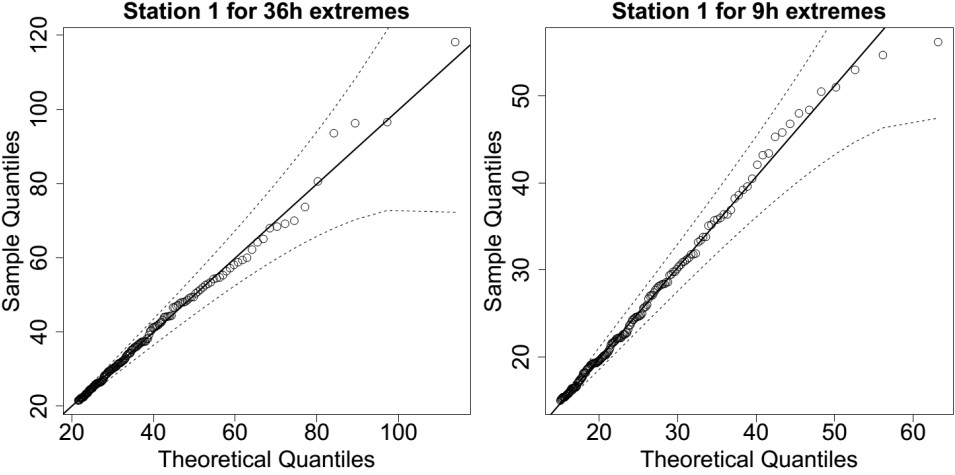

**Figure 7.** QQ plots for the fitted GPD at one representative station, dotted lines are the 95% confidence bounds, and the

solid diagonal line indicates a perfect fit.

The inverted max-stable process across different durations was calibrated to determine dependence

parameters. The theoretical pairwise residual tail dependence coefficient function between two

locations ($x_1$ and $x_2$) was calculated based on Eq. (3) and Eq. (4), and the observed pairwise residual

tail dependence coefficient $\eta$ was calculated using Eq. (2). The model has a reasonable fit to the

observed data given the small number of dependence parameters. Figure 8 shows the pairwise residual

tail dependence coefficients for the Brown-Resnick inverted max-stable process versus distance. The

black points are the observed pairwise residual tail dependence coefficients, while the red lines are the

fitted pairwise residual tail dependence coefficient functions. A coefficient equal to 1 indicates

complete spatial dependence, and a value of 0.5 indicates complete spatial independence. The top-left

panel shows the dependence between 36 hr extremes across space, with the distance $h = 0$

corresponding to "complete dependence". It also shows the dependence decreasing with increasing

distance.

The remaining panels of Fig. 8 show the dependence of 36 vs. 9 hr extremes, 36 vs. 6 hr extremes,

and 36 vs. 3 hr extremes, with the latter two duration combinations not being used directly in the

study but nonetheless showing the model performance across several durations. As expected, the

dependence levels are weaker compared with 36 vs. 36 hr extremes at the same distance, especially at



the distance of 0. This is expected, as the dependence at the same site between annual maxima at
different durations will be lower than between annual maxima at the same duration. This is because
the annual maxima of different durations may arise from different storm events (Zheng et al., 2015).

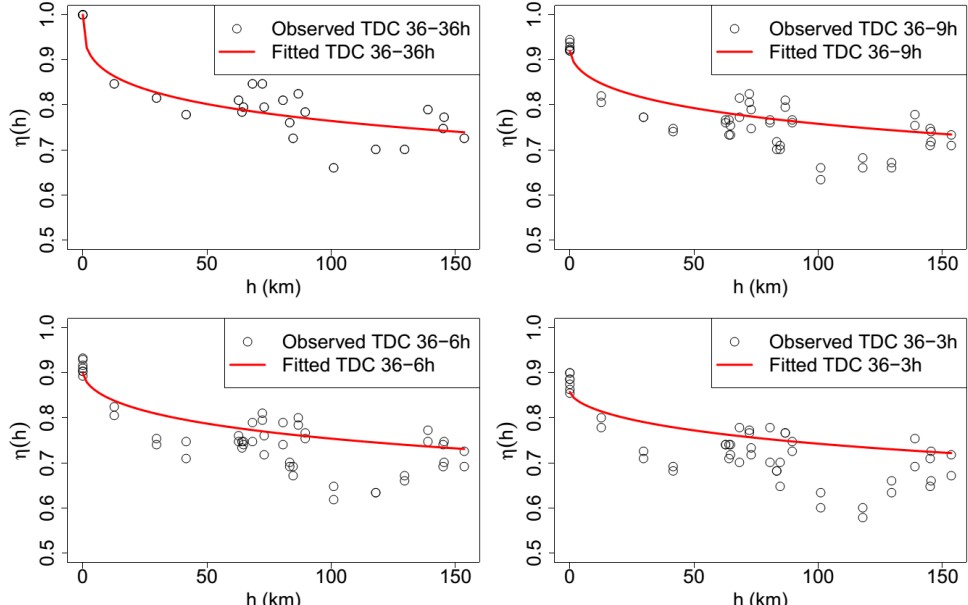


**Figure 8.** Plots of pairwise residual tail dependence coefficient (TDC) against distance for 36 hr extremes and 36 hr
extremes (left), and for 36 hr extremes and 9 hr extremes (right). The black points are estimated residual tail dependence
coefficients (TDC) for pairs of sub-daily stations, and the red lines are theoretical residual tail dependence coefficient (TDC)
function.
***5.2. Estimating conditional rainfall extremes and corresponding conditional flows for evacuation***
***route design***
The recommended approach for estimating conditional rainfall extremes is demonstrated by
considering a hypothetical evacuation route across location $x_2$, given a flood occurs at location $x_1$,
evaluated using Eq. (B.3). This approach is applied to a case study of the Pacific Highway upgrade
project that contains five main river crossings (from Fig. 3). For evacuation purposes, we need to
know "what is the probability that a bridge fails only once on average every $M$ times (e.g., $M = 10$
for a 10-year event) that its neighbouring bridge is flooded?" This section provides the conditional





estimates for two pairs of neighbouring bridges in the case study that have the shortest Euclidean
distances, i.e. pairs $(x_1, x_2)$ and $(x_2, x_3)$. The comparisons of unconditional and conditional maps are
given in Fig. 9 and Fig. 10, and the corresponding unconditional and conditional flows are given in
Fig. 11.
The left panel of Fig. 9 provides the pointwise 10-year unconditional return level map over the case
study area for 36 hr rainfall extremes. The value at the location of interest—the blue star (the centroid
of Bellinger catchment)—is around 260 mm. The right panel of Fig. 9 indicates that when accounting
for the effect of a 20-year event for 36 hr rainfall extremes happening at the location of the red star
(the centroid of Kalang River catchment), the pointwise 10-year conditional return level at the blue
star rises to around 453 mm (i.e., 1.74 times the unconditional value).

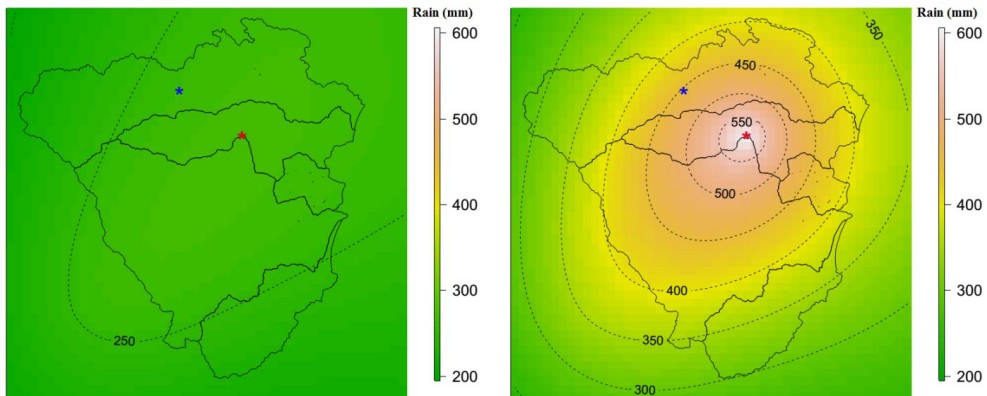


**Figure 9.** Pointwise 10-year unconditional return level map (mm) for 36 hr extremes (left), and pointwise 10-year
conditional return level map (mm) for 36 hr extremes given a 20-year event for 36 hr extremes happen at location of the red
star for the centroid of Kalang River catchment (right). The colour scales are the same for comparison.
Figure 10 provides similar plots to Fig. 9 for another pair of locations having different durations of
rainfall extremes due to different times of concentration in each catchment. Here, the location of
interest is the centroid of the Deep Creek catchment (the blue star in Fig. 10) and the conditional point
is the centroid of the Kalang River catchment (the red star in Fig. 10). The pointwise 10-year
unconditional and conditional return levels at the location of the blue star are 134 mm and 194 mm,
respectively. The relative difference between the conditional and unconditional return levels is only





1.45 times, compared with 1.74 times for the case in Fig. 9. This is because the pair of locations in
Fig. 10 has a longer distance than those in Fig. 9, so that the dependence level is weaker. Moreover,
the location pair in Fig. 10 was analysed for different durations (between 36 and 9 hr extremes),
which has weaker dependence than the case of the equivalent durations in Fig. 9 (between 36 and 36
hr), based on Fig. 8.

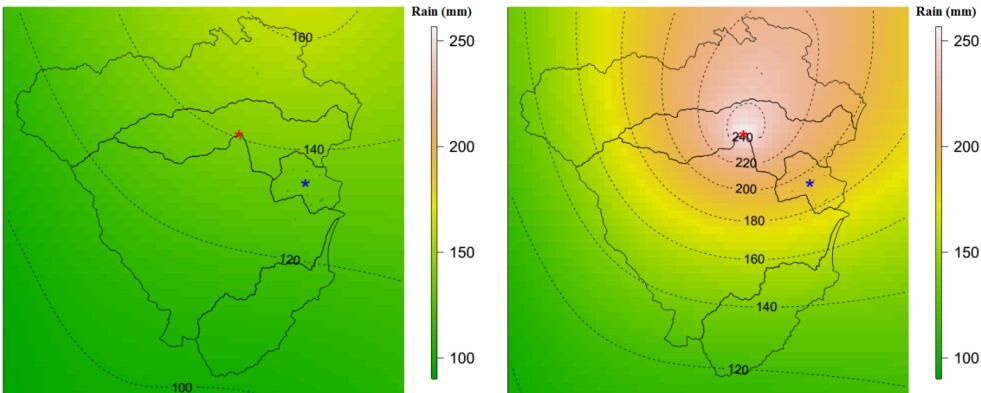


**Figure 10.** Pointwise 10-year unconditional return level map (mm) for 9 hr extremes (left), and pointwise 10-year

conditional return level map (mm) for 9 hr extremes, given a 20-year event for 36 hr extremes happens at location of the red

star for the centroid of the Kalang River catchment (right). The colour scales are the same for comparison.

The unconditional and conditional return levels are transformed to flood flows via the hydrological
model WBNM previously calibrated to each catchment (WMAWater, 2011). The unconditional and
conditional return levels were extracted at the centroid of each main catchment, which were then
converted to the average spatial rainfall using an areal reduction factor (ARF). The corresponding
unconditional and conditional flood flows at the river crossing in the Bellinger catchment
(corresponding to the unconditional and conditional rainfall extremes in Fig. 9) are given in Fig. 11
(left panel). Similar plots for the river crossing in the Deep Creek catchment (corresponding to the
unconditional and conditional rainfall extremes in Fig. 10) are given in Fig. 11 (right panel).





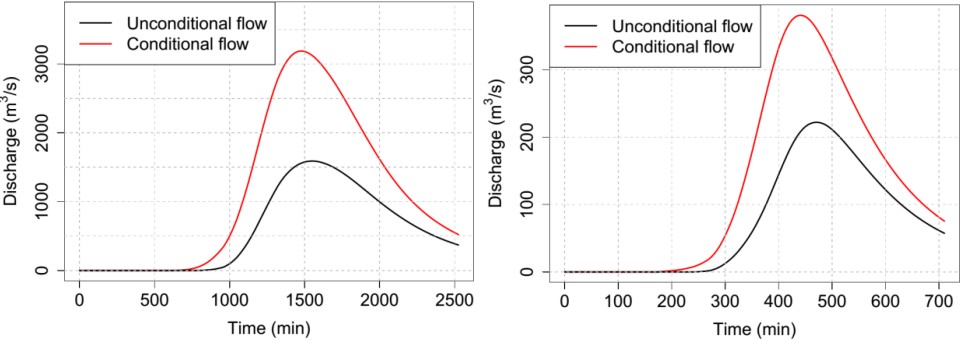


**Figure 11.** Comparison between conditional flows (red line) and unconditional flows (black line). (left) At the river crossing

in the Bellinger catchment: conditional flow caused by a 10 year conditional event for 36 hr rainfall in considering the effect

of a 20 year event for 36 hr rainfall occurring at the river crossing in the Kalang River catchment, and unconditional flow

caused by a 10 year unconditional event for 36 hr. (right) At the river crossing in the Deep Creek catchment: conditional

flow caused by a 10 year conditional event for 9 hr rainfall in considering the effect of a 20 year event for 36 hr rainfall

occurring at the river crossing in the Kalang River catchment, and unconditional flow caused by a 10 year unconditional

event for 9 hr rainfall.

The left panel of Fig. 11 indicates that the peak conditional flow at the river crossing in the Bellinger

catchment is almost 2.0 times higher than that for unconditional flow. The time taken to reach to the

peaks is the same for both cases. This is because this river crossing is affected by a large region with a

long time of concentration (36 hr); the impact of rainfall losses on the hydrograph is insignificant.

This difference is a direct result of the conditional relationship being more stringent than the

unconditional relationship. Given that there is an existing extreme event nearby, it is more likely for

an extreme event to occur at another location of interest in the region. If a bridge design were to take

into account this extra criterion for the purposes of evacuation planning it would require the design to

be at a higher level.

Shown in the right panel in Fig. 11, the peak of the conditional flow at the river crossing in the Deep

Creek catchment occurred earlier, and is around 1.7 times higher than that for the unconditional flow.

This is due to the fact that the river crossing in Deep Creek covers a small region with a short time of

concentration (9 hr) and the impact of rainfall losses on the hydrograph is significant.





Although Fig. 11 shows a difference in terms of the time taken to reach the peak flows, the two design
hydrographs are separate and this is not a physical timing difference. The relevant feature of the
conditional design hydrograph is the peak, and timing information is not a part of the method.
The difference between the maximum discharge of conditional and unconditional flows at the river
crossing in the Bellinger catchment is shown in Fig. 12 for the case of a 20-year event occurring in the
Kalang River catchment nearby. The relationship with AEP indicates that the difference between the
maximum discharge of conditional and unconditional flows decreases when AEP increases, and that
the difference approaches zero when the AEP increases to above 50% (i.e. a 2-year return period).

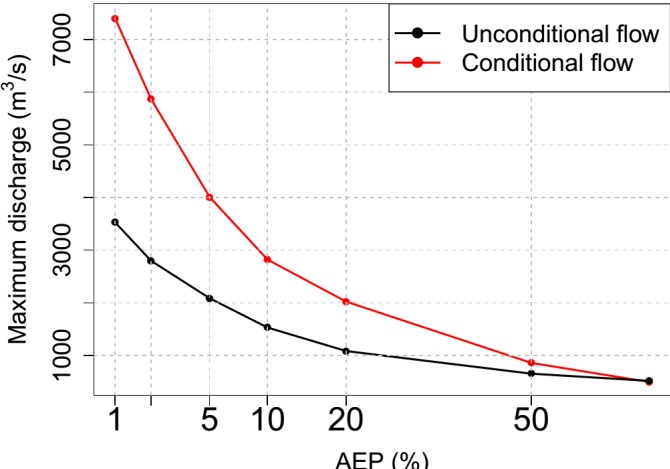


**Figure 12.** Plot for peak of conditional flow (red points) caused by conditional flood-producing rainfall and peak of
unconditional flow (black points) for different annual exceedance probabilities (AEP) at the river crossing in the Bellinger
catchment. This plot considers the effect of a 20-year event occurring at the river crossing in the Kalang River catchment.
*5.3. Estimating the failure probability of the highway section based on the joint probability of*
*rainfall extremes*
The recommended approach for estimating the overall failure probability of a system is demonstrated
by considering a hypothetical traffic system with multiple river crossings at locations $x_1, \ldots, x_N$. If
there is a one-to-one correspondence between extreme rainfall intensity and flood magnitude, the
overall failure probability will be approximately equal to the probability that there is at least one river
crossing whose contributing catchment has rainfall extremes exceeding the design level, which can be





estimated using a large number of simulations from the spatial rainfall model. This approach is
applied to the Pacific Highway upgrade project containing five river crossings. A set of 10,000 year
simulated rainfall (Section 4.5) is generated from the fitted model (Section 5.1) to calculate the overall
failure probability of the highway section. This process is repeated 100 times to estimate the average
failure probability, under the assumption that all river crossings are designed to the same individual
failure probability.
Figure 13 is a plot of the overall failure probability of the highway and the failure probability of each
individual river crossing (black). Similar relationships for the cases of complete dependence (blue)
and complete independence (red) are also provided for comparison. For the case of complete
dependence, when the whole region is extreme at the same time, the overall failure probability of the
highway is equal to the individual river crossing failure probability and it represents the best case. The
worst case is complete independence where extremes do not happen together unless by random
chance; this means the failure probability of the highway is much higher than that for individual river
crossings. Taking into account the real dependence, there are some extremes that align and it seems
from the Fig. 13 that this is a relatively weak effect. As an example from Fig. 13, to design the
highway with a failure probability of 1% annual exceedance probability (AEP), we would have to
design each individual river crossing to a much rarer AEP of 0.25% (see green lines in Fig. 13).






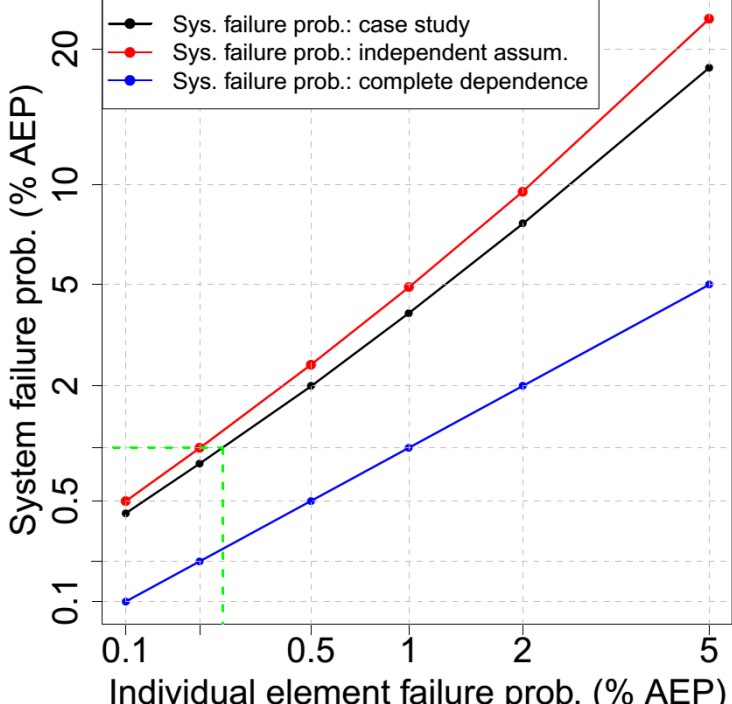

**Figure 13.** Relationship between system failure probability and individual element failure probability in % annual

exceedance probability (% AEP). The black colour is for the case study, the red colour is for the case of complete

independence, and the blue is for the case of complete dependence.

## 6. Discussion and Conclusions

Hydrological design has conventionally focussed on individual catchments and individual extremes.

Such an approach can lead to an underestimation of wider system risk of flooding since weather

systems exhibit dependence in space and time, which can lead to the coincidence of extremes. A

number of methods have been developed to address the problem of antecedent moisture within a

single catchment, by accounting for the temporal dependence of rainfall at locations of interest

through loss parameters or sampling rainfall patterns (Rahman et al., 2002). However, there have been

fewer methods that account for the spatial dependence of rainfall across multiple catchments, due in

part to the complexity of representing the effects of spatial dependence in risk calculations. Different





catchments can have different times of concentration, so spatial dependence may also imply the need
to consider dependence across different durations of extreme rainfall bursts.
Recent and ongoing advances in modelling spatial rainfall extremes provide an opportunity to revisit
the scope of hydrological design. Such models include a max-stable model fitted using a Bayesian
hierarchical approach (Stephenson et al., 2016), max-stable and inverted max-stable models (Nicolet
et al., 2017; Padoan et al., 2010; Russell et al., 2016; Thibaud et al., 2013; Westra and Sisson, 2011)
and latent-variable Gaussian models (Bennett et al., 2016b). The ability to simulate rainfall over a
region means that hydrological problems need not be confined to individual catchments, but may
cover multiple catchments. Civil infrastructure systems such as highways, railways or levees are such
examples, since the failure of any one element may lead to overall failure of the system. Alternatively,
where there is a network, the failure of one element may have implications for the overall system to
accommodate the loss, by considering alternative routes. With models of spatial dependence and
duration dependence of extremes there is a new and improved ability to address these problems
explicitly as part of the design methodology.
This paper demonstrated an application for evaluating conditional and joint probabilities of flood at
different locations. This was achieved with two examples: (i) the design of a river crossing that will
fail once on average every $M$ times given that its neighbouring river crossing is flooded; and (ii)
estimating the probability that a highway section, which contains multiple river crossings, will fail
based on the failure probability of each individual river crossing. Due to the lack of continuous
streamflow data and subdaily limitations of rain-based continuous simulation, this study used an
event-based method of conditional and joint rainfall extremes to estimate the corresponding
conditional and joint flood flows. The spatial rainfall was simulated using an asymptotically
independent model, which was then used to estimate conditional and joint rainfall extremes. An
empirical method was obtained from the framework of Le et al. (2018b) to make an asymptotically
independent model—the inverted max-stable process—able to capture the spatial dependence of
rainfall extremes across different durations. The fitted residual tail dependence coefficient function
showed that the model can capture the dependence for different pairs of durations. For our example,



the highest ratio of conditional to unconditional extremes was 1.74, for the two catchments having the strongest dependence (Fig. 9). The corresponding conditional flows were then estimated using a hydrological model WBNM and shown to be strongly related to the ratio of conditional and unconditional rainfall extremes (Fig. 11).

The joint probability of rainfall extremes for all catchments and for all possible pairs of catchments in the case study area was estimated empirically from a set of 10,000 years of simulated rainfall extremes, repeated 100 times to estimate the average value. The results showed that there were differences in the failure probability of the highway after taking into account the rainfall dependence, but the effect was not as emphatic as with the case of conditional probabilities. The difference in the failure probability became weaker as the return period increased, which is consistent with the characteristic of asymptotically independent data (Ledford and Tawn, 1996; Wadsworth and Tawn, 2012). A relationship was demonstrated (Fig. 13) to show how the design of the overall system to a given failure probability requires the design of each individual river crossing to a rarer extremal level than when each crossing is considered in isolation. For the case study example, it would be necessary to design each bridge to a 0.25% AEP event in order to obtain a system failure probability of 1%.

There is a need to reimagine the role of intensity-duration-frequency curves. Conventionally they have been developed as maps of the marginal rainfall in a point-wise manner for all locations and for a range of frequencies and durations. The increasing sophistication of mathematical models for extremes, computational power and interactive graphics abilities of online mapping platforms means that analysis of hydrological extremes could significantly expand in scope. With an underlying model of spatial and duration dependence between the extremes, it is not difficult to conceive of digital maps that dynamically transform from the marginal representation of extremes to the corresponding representation conditional extremes after any number of conditions are applied. This transformation is exemplified by the differences between left and right panels in Fig. 9 and Fig. 10. Enhanced IDF maps would enable a very different paradigm of design flood risk estimation, breaking away from analysing individual system elements in isolation to emphasize the behaviour of entire system.




**Appendix A. Calculation of empirical tail dependence coefficient**

To illustrate how Eq. (2) in the manuscript is calculated, consider a set of $n = 10$ observed values at the two locations: $Z_1 = c(5,9,1,2,10,3,8,6,4,7)$; $Z_2 = c(10,1,7,6,4,3,9,2,8,5)$. First, $Z_1$ and $Z_2$ are converted to empirical cumulative probability estimates via the Weibull plotting position formula $P = j/(n + 1)$ where $j$ is ranked index of a data point giving $P_1 = c(0.455, 0.818, 0.091, 0.182, 0.909, 0.273, 0.727, 0.545, 0.364, 0.636)$ and $P_2 = c(0.909, 0.091, 0.636, 0.545, 0.364, 0.273, 0.818, 0.182, 0.727, 0.455)$. Assume that interest is in values above a threshold $u = 0.5$, in other words, $P\{Z_2 > z\} = P\{P_2 > u\} = 0.5$. In this case we have only one pair, at the index of 7, that satisfy both $P_1$ and $P_2$ are greater than $u = 0.5$, thus $P\{Z_1 > z, Z_2 > z\} = P\{P_1 > u, P_2 > u\} = 1/10 = 0.1$. The calculation of the empirical tail dependence coefficient is then

$$\eta(x_1, x_2) = \frac{\log P\{Z_2 > z\}}{\log P\{Z_1 > z, Z_2 > z\}} = \frac{\log P\{P_2 > u\}}{\log P\{P_1 > u, P_2 > u\}} = \frac{\log(0.5)}{\log(0.1)} = 0.301. \qquad (A.1)$$

**Appendix B. Equations for bivariate conditional and joint probabilities for inverted max-stable**

In the context of this study, the conditional probability $P\{Z_2 > z_2 | Z_1 > z_1\}$ is obtained from the bivariate inverted max-stable process cumulative distribution function (CDF) in unit Fréchet margins (Thibaud et al., 2013), which is given as:

$$P\{Z_1 \leq z_1, Z_2 \leq z_2\} = 1 - \exp\left\{-\frac{1}{g_1}\right\} - \exp\left\{-\frac{1}{g_2}\right\} + \exp[-V\{g_1, g_2\}], \qquad (B.1)$$

where $g_1 = -1/\log\{1 - \exp(-1/z_1)\}$, $g_2 = -1/\log\{1 - \exp(-1/z_2)\}$, and the exponent measure $V$ (Padoan et al., 2010) is defined as:

$$V\{g_1, g_2\} = -\frac{1}{g_1}\Phi\left\{\frac{a}{2} + \frac{1}{a}\log\frac{g_2}{g_1}\right\} - \frac{1}{g_2}\Phi\left\{\frac{a}{2} + \frac{1}{a}\log\frac{g_1}{g_2}\right\}. \qquad (B.2)$$

In Eq. (B.2), $\Phi$ is the standard normal cumulative distribution function, $a = \sqrt{2\gamma_{ad.}(h)}$ with $\gamma_{ad.}(h)$ is the variograms that was mentioned in the explanation of Eq. (4) in the manuscript.

In unit Fréchet margins, the relationship between the return level $z$ and the return period $T$ is given as $z = -1/\log(1 - 1/T)$, and the conditional probability for the max-stable process can then be estimated using:

$$P\{Z_2 > z_2 | Z_1 > z_1\} = T_1\left[\frac{1}{T_1} - \exp\left(-\frac{1}{z_2}\right) + P\{Z_1 \leq z_1, Z_2 \leq z_2\}\right], \qquad (B.3)$$



where $T_1$ is the return period corresponding to the return level $z_1$.

**Acknowledgments**

The lead author was supported by the Australia Awards Scholarships (AAS) from Australia
Government. A/Prof Westra was supported by Australian Research Council Discovery grant
DP150100411. We thank Mark Babister and Isabelle Testoni of WMA Water for providing the
hydrologic models for the case study; and Leticia Mooney for her editorial help in improving this
manuscript. The rainfall data used in this study were provided by the Australian Bureau of
Meteorology, and can be obtained from the corresponding author.





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
