# Peer review of "Spatially dependent flood probabilities to support the design of civil infrastructure systems"

_Hydrology and Earth System Sciences, 2018_

## Referee Comment (RC1) · J. Beckers (Referee) · 29 Sep 2018

This manuscript describes the application of a correlation model for spatially dependent rainfall and hydrological response of four subcatchments that can cause flooding of a highway. The road is blocked if either of flows from the four subcatchments exceeds a critical threshold. The probability of system failure (road blockage) thus depends on the exceedance probability of four thresholds by four correlated stochastic variables.

Although the scientific methods that are used in this study may not be entirely new, the explanation of spatial dependency of rainfall and application to a practical case study are very clear and a pleasure to read. After reading this manuscript, a decision maker

should understand that it is important to take this correlation into account.

I have only one specific comment: the core of the technical approach I would consider to be the correlation model, i.e. the Brown-Resnick inverted max-stable process. This method is not explained at all. Instead, the authors choose to refer to literature. Although a fully detailed description of the B-R algorithm may be too much, it would be good if the essence of this method is explained briefly.

---

## Author Comment (AC1) · 1 Oct 2018

Thank you very much for your suggestion. Although a full explanation of the B-R model is very long and technical and well-covered in other papers, we will provide a brief summary of the main technique through the inclusion of a high-level algorithm in the next version of the manuscript.

---

## Referee Comment (RC2) · Anonymous Referee #2 · 7 Nov 2018

Summary: The manuscript describes a statistical framework based on an inverted max-stable process allowing to account for the spatial dependence of rainfall across durations. Application is made for a case study in New South Wales, Australia. Using the proposed framework, the author are able to compute conditional and joint return levels of rainfall. Through the use of rainfall ARFs and of an hydrological model, that authors also derive conditional and joint return levels of river flows. Finally the authors derive the failure probability of a highway section, defined as the probability that flood magnitude at any of the five river crossings exceeds a given threshold, assuming a 1-1 correspondence between flood magnitude and rainfall over a catchment.

[Figure]

Main comments: The article is well written and mainly clear. The two risk applications of Section 5.1 and 5.2 are very interesting, particularly 5.2 (failure probability of a highway section) which seems to me to be more related to "real" issues than 5.1. The subject is absolutely worth publishing in HESS. However I raise below a couple of major issues to be addressed before publication:

The use of "Intensity-Duration-Frequency curves" in the title seems at the moment misleading. I would have expected from this expression to see e.g. joint or conditional IDF curves at a given station/catchment, i.e. the IF curves for several durations. Here actually only one duration is used for every catchment – basically the concentration time of the catchment. So I'd be tempted to replaced "IDF" in the title (and the text) by "return levels".

I'm puzzled about the GPD fits. If I understood correctly, GPD are fitted to 9 and 36 hr rainfall exceedances. If moving windows are considered, then there is a very strong auto-correlation for both the 9 and 36 hr rainfall values. Have you taken this into account in the fits? A declustering method should be applied. This may be the reason why the fits for 36 hr extremes are usually poorer than for 9 hr extremes (see Figs S5 and S6).

The part regarding the ARFs seems obscure to me (Section 4.5). Basically I isn't clear tome what the ARF allow for. I interpret between the lines that they allow to transform point return levels to spatial return levels over a catchment. However the way ARFs are described is very confusing to me. For example l. 346 states that "the rainfall extremal estimates need to be converted to the average spatial rainfall using an ARF". First I don't understand what are the "rainfall estimates" (rainfall return levels?). Second I guess that " average spatial rainfall" should be "spatial rainfall return levels". I recommend clarifying Section 4.5 and part of the Introduction dealing with ARFs.

Expressions such as "10-year conditional return level map given a 20-year event happen at x" are confusing to me. Wouldn't it be less confusing to say this is the levels

expected to occur on average once every 3650 times when a 20-year event happen at x. The "10-year" is misleading to me in that case due to the conditioning.

I'm confused with the reference to "annual maxima", whereas the article considers peaks-over-threshold. For example Fig 1 illustrates the case of annual maxima (GEV), which is not the case here. L. 421-423 talks about annual maxima instead of exceedances.

I haven't understood what is the AEP of Fig 12 and 13. I guess it would be clearer to replace AEP by return periods.

Minor comments: l. 111: Le et al → no brackets

l. 113 AFR → ARF

l. 116-117: I may be clearer to exemplify (i) in terms of evacuation route design as you do in Section 5.1.

Fig. 3: add the station numbers 1, 2, 3...

Fig. 4 estimate conditional rainfall → estimate conditional probability rainfall

l. 277: where → to be removed

l. 294-296: why don't you estimate all parameters (beta, q, c) together?

l. 333-334 it is also noted .. 9 hrs → is it useful here?

Section 4.5: to be rewritten to clarify the ARFs as said above

l. 346: rainfall estimates: what are they?

l. 353-354: the BR process → for what duration? With which parameters?

l. 360: empirical distributions → I'm confused here. If you use empirical distributions below the threshold, how can you have rainfall at ungauged sites (maps)?

l. 373: multiple durations → Is the algorithm of Dombry still applicable in this case? I'm

not sure to see how it works for multiple durations.

l. 373 in this case... pair of locations → I don't understand it at all. What covariance matrix are you talking about?

l. 378 rainfall hyetographs → what rainfall are you talking about? Spatial rainfall over the catchments?

Fig. 6: is it useful here? It could be in the supplementary material.

l. 385 & 387: hydrological models → hydrological model layouts

l. 398: did you apply declustering before estimating the GPDs?

Fig. 7 and SM: there is a huge difference between the extremes at the different stations, e.g. station 2 vs station 6. Could you comment on this? Also what method did you use to produce the confidence bands?

l. 421-423: I'm lost here. Do you fit the BR process to annual maxima or exceedances?

Caption of Fig 8: Abbreviation TDC is useless

Fig. 9: I don't understand how you get the maps. For this you need the marginal distribution of rainfall at every pixel. How do you get this?

l. 469: average spatial rainfall: I'm confused. How can you transform return levels to averages?

Fig. 11 at the river crossing: which crossing are you talking about? There are several.

l. 495-497: Although Fig 11 . . . not part of the method → I don't understand these two sentences. What do you mean by "this is not a physical timing difference"?

Fig. 12: I don't understand the AEP. Wouldn't it be clearer with return periods instead of AEP?

l. 511: extreme rainfall intensity → over a catchment?

[Figure]

l. 520: and → as a function of?

Fig. 13: as for Fig. 12, would be clearer to show return periods in the x-axis?

Caption of Fig. 13: please explain what are the green segments

l. 529: 1% annual exceedance prob → 1% AEP

l. 573: 1.74 → I guess this number depends on the considered levels

l. 611: inverted max-stable → inverted max-stable process

Fig. S1: I don't understand the figure. Could you please explain what a given point represents? Given Table 1, I would have expected to have points at A=91, 294, 341, 771, 1020, which is not the case.

―――――――――――――――――――――

---

## Referee Comment (RC3) · Anonymous Referee #3 · 10 Dec 2018

In general, the paper is well written. However, I have some concerns regarding the real contribution (novelty), connection with the literature and in particular with copula studies, as well as comparison with other models. Main comments: 1. Some important papers related to the topic are missing and more importantly the comparison with them not only in terms of results but also in terms of advantages and drawbacks (e.g. Bardossy and Pegram, 2009, Durocher et al. 2016 and Requena et al. 2018). 2. Regarding the issues motivating the study: the first one seems to be already fixed by Le et al. 2018b (as indicated on page 5), and the second issue is not clear (seems to be written as a statement not as an issue). 3. The topic can also be closely related to regional frequency analysis or estimation at ungauged basins. The authors did not make

this connection or show the difference. In the first case (similarity or connection), a huge literature exists and should be considered. 4. The paper focused on a case study (a given set of data). However, the effect of some factors on the performance of the model as not discussed and not studied: for instance, and not limited to, the dimensionality (number of sites) and the size of the subgroups. 5. An important missing element from the paper is the notion of copulas which is the most important when dealing with dependence. There is a huge literature in both hydrology and statistics (even in spatial dependence). I'm surprised to not see it in the paper. 6. In section 4: why the GPD is used directly without model selection procedure? Why it is the same for all sites? The GPD is usually asymptotically justified which is not enough (and less justified in hydrology because of the sample sizes) and does not depend on the data at hand. It should be considered as a distribution among others (like GEV for block extremes). 7. Lines 245-248: please provide other alternative models and justify the choice of your model. 8. The assumption, on page 11 line 215, is it reasonable? Is it verified in your case study? 9. How the hydrological model (ex. WBNM) is integrated in the steps of fig 4? Other comments: 1. Fig 4: Why in the independent model, no fitting is required? What it means? 2. Sentence from lines 237-240 is long and not clear. Please consider reformulating. 3. Page 13: this text requires to be more accurate about the terms and notation. 4. Lines 287-290: is this case not covered by equation 4? 5. All text in page 16 and part of page 17 seems trivial and does not worth all this space. Other more important information deserve this space. 6. It is not clear in section 4.6 if the authors consider one hydrological model (WBNM) or other models (see for instance lines 376 and 384). 7. Line 408 : how you can say the model has reasonable fit? Based on what? And compared to what? 8. Line 538 : I'm not sure about this statement. It is not true in many situations.

Durocher, M., Chebana, F., & Ouarda, T. B. (2016). On the prediction of extreme flood quantiles at ungauged locations with spatial copula. Journal of Hydrology, 533, 523-532.

Bárdossy, A., & Pegram, G. G. S. (2009). Copula based multisite model for daily precipitation simulation. Hydrology and Earth System Sciences, 13(12), 2299-2314. Requena, A. I., Chebana, F., & Ouarda, T. B. (2018). A functional framework for flow-duration-curve and daily streamflow estimation at ungauged sites. Advances in Water Resources, 113, 328-340.

---

## Author Comment (AC3) · 21 Jan 2019

**Reference Code:** hess-2018-393

**Title:** Spatially dependent Intensity-Duration-Frequency curves to support the design of civil infrastructure systems

**Corresponding Author:** Phuong Dong Le (The University of Adelaide)

**Contributing Authors:** Michael Leonard and Seth Westra

**Response to the Reviewer #3**

*In general, the paper is well written. However, I have some concerns regarding the real contribution (novelty), connection with the literature and in particular with copula studies, as well as comparison with other models. Main comments:*

**Response:** Thank you for your comments. We respond in detail below (your comments in italic font and our responses in normal font).

**Major comment #1:**

*1. Some important papers related to the topic are missing and more importantly the comparison with them not only in terms of results but also in terms of advantages and drawbacks (e.g. Bardossy and Pegram, 2009, Durocher et al. 2016 and Requena et al. 2018).*

**Response**: Thank you for the suggestion. We will add discussion on these paper to the revised manuscript.

**Major comment #2:**

*2. Regarding the issues motivating the study: the first one seems to be already fixed by Le et al. 2018b (as indicated on page 5), and the second issue is not clear (seems to be written as a statement not as an issue).*

**Response**: Thank you for pointing this out. We will rewrite this section to make it clearer. The second issue relates to the spatial properties of asymptotic dependence (explored in Le et al., 2018a). While these two issues have been separately addressed in previous papers, the contribution is to show how to combine the methods to solve a realistic design problem.

References used for this response:

Le, P. D., Davison, A. C., Engelke, S., Leonard, M., and Westra, S.: Dependence properties of spatial rainfall extremes and areal reduction factors, Journal of Hydrology, Submitted, 2018a.

Le, P. D., Leonard, M., and Westra, S.: Modeling Spatial Dependence of Rainfall Extremes Across Multiple Durations, Water Resources Research, 54, 2233-2248, 2018b.

**Major comment #3:**

*3. The topic can also be closely related to regional frequency analysis or estimation at ungauged basins. The authors did not make this connection or show the difference. In the first case (similarity or connection), a huge literature exists and should be considered.*

**Response**: Thanks for your comment. We will discuss differences to regional frequency analysis and methods of estimation in the revised manuscript.

**Major comment #4:**

*4. The paper focused on a case study (a given set of data). However, the effect of some factors on the performance of the model as not discussed and not studied: for instance, and not limited to, the dimensionality (number of sites) and the size of the subgroups.*

**Response**:

Thanks for your comment. We will review the methodology more closely to include additional details on factors of the model performance, including discussion on the effects of additional sites.

**Major comment #5:**

*5. An important missing element from the paper is the notion of copulas which is the most important when dealing with dependence. There is a huge literature in both hydrology and statistics (even in spatial dependence). I'm surprised to not see it in the paper.*

**Response**: We will add literature on copulas into the revised manuscript.

**Major comment #6:**

*6. In section 4: why the GPD is used directly without model selection procedure? Why it is the same for all sites? The GPD is usually asymptotically justified which is not enough (and less justified in hydrology because of the sample sizes) and does not depend on the data at hand. It should be considered as a distribution among others (like GEV for block extremes).*

**Response**: Thank you for this comment. We used the GPD because, in contrast to block maxima, it allows us to consider concurrent rainfall extremes and therefore enables the study of dependence. The intention in this paper is not to work through repetitive fitting of different distributions, but to demonstrate a plausible method based on joint rainfall extremes for the design of linear infrastructure. The same distribution is used at each site with variation at each site carried by the parameters. The marginal model adopted is not perfect, but it is plausible, and sufficient for the intent of showing the application of rainfall dependence to design.

**Major comment #7:**

*7. Lines 245-248: please provide other alternative models and justify the choice of your model.*

**Response**: Thank you. We will add justification of the choice of the Brown-Resnick model in the revised manuscript. For example, Le et al. (2018a) show it has better performance than the extremal-t model.

Le, P. D., Davison, A. C., Engelke, S., Leonard, M., and Westra, S.: Dependence properties of spatial rainfall extremes and areal reduction factors, Journal of Hydrology, Submitted, 2018a.

**Major comment #8:**

*8. The assumption, on page 11 line 215, is it reasonable? Is it verified in your case study?*

**Response**: Thank you very much. The assumption of AEP neutrality in rainfall-runoff design is a standard assumption when using IDF curves. While the assumption is in widespread use, it is not without limitation and we will provide brief discussion and reference to two papers that explore this issue.

Bennett, B., Leonard, M., Deng, Y., & Westra, S. (2018). An empirical investigation into the effect of antecedent precipitation on flood volume. Journal of Hydrology, 567, 435-445.

Rahman, A., Weinmann, P. E., Hoang, T. M. T., & Laurenson, E. M. (2002). Monte Carlo simulation of flood frequency curves from rainfall. Journal of Hydrology, 256(3-4), 196-210.

**Major comment #9:**

*9. How the hydrological model (ex. WBNM) is integrated in the steps of fig 4?*

**Response**: The hydrological model (i.e. WBNM) is used to transform the conditional rainfall to conditional flow. A label will be added in the revised version of the manuscript to show this (on the arrow between the see the squares for Section 4.5 and Section 4.6 in the top-right of Figure 4).

**Minor comment #1:**

*1. Fig 4: Why in the independent model, no fitting is required? What it means?*

**Response**: Thank you for pointing this out. The term "the independent model" here is not clear. We will change it to "the case of independence" and will clarify that we mean the case where rainfall extremes occur independently in space.

**Minor comment #2:**

*2. Sentence from lines 237-240 is long and not clear. Please consider reformulating.*

**Response**: Thank you. We will reword these sentences in the revised manuscript.

**Minor comment #3:**

*3. Page 13: this text requires to be more accurate about the terms and notation.*

**Response**: Thank you very much. We will clarify this text in the revised manuscript.

**Minor comment #4:**

*4. Lines 287-290: is this case not covered by equation 4?*

**Response**: Thank you. We will rewrite this comment on equation 4. We will clarify that the equation can be used for both cases, but that difference parameters will be required, since the dependence of a short duration extreme given a longer duration extreme is not the same as the dependence of a long duration extreme given a short duration.

**Minor comment #5:**

*5. All text in page 16 and part of page 17 seems trivial and does not worth all this space. Other more important information deserve this space.*

**Response**: We will put remove this material and put it as supplementary, which will create significantly more space. We would like to leave it in supplementary material rather than remove it outright because some readers may prefer the straightforward explanation given the practical design focus in the paper.

**Minor comment #6:**

*6. It is not clear in section 4.6 if the authors consider one hydrological model (WBNM) or other models (see for instance lines 376 and 384).*

**Response**: Thank you for your comment. There is only one type of model (WBNM), but different configurations for each catchment. We will clarify this in the revised text.

**Minor comment #7:**

*7. Line 408 : how you can say the model has reasonable fit? Based on what? And compared to what?*

**Response**: Thank you. We will more explicitly indicate that the comment on fitting relates to Figure 8. We will also emphasize that the main feature of the model shown in these figures is the relationship at $h=0$, for the case of dependence between two different durations at the same location.

**Minor comment #8:**

*8. Line 538 : I'm not sure about this statement. It is not true in many situations.*

**Response**: Thank you for your comment. We will restrict our commentary to conventional hydrological design that is based on IDF curves, which is more defensible than the original comment which was too general. By construction IDF curves are focused are point-wise estimators of extremes, thus a given design is focused on independent application of univariate statistics.

---

## Author Response (AR1)

**Reference Code:** hess-2018-393

**Title:** Spatially dependent Intensity-Duration-Frequency curves to support the design of civil infrastructure systems

**Corresponding Author:** Phuong Dong Le (The University of Adelaide)

**Contributing Authors:** Michael Leonard and Seth Westra

**Response to the Reviewer #1**

*This manuscript describes the application of a correlation model for spatially dependent rainfall and hydrological response of four subcatchments that can cause flooding of a highway. The road is blocked if either of flows from the four subcatchments exceeds a critical threshold. The probability of system failure (road blockage) thus depends on the exceedance probability of four thresholds by four correlated stochastic variables.*

*Although the scientific methods that are used in this study may not be entirely new, the explanation of spatial dependency of rainfall and application to a practical case study are very clear and a pleasure to read. After reading this manuscript, a decision maker should understand that it is important to take this correlation into account.*

*I have only one specific comment: the core of the technical approach I would consider to be the correlation model, i.e. the Brown-Resnick inverted max-stable process. This method is not explained at all. Instead, the authors choose to refer to literature. Although a fully detailed description of the B-R algorithm may be too much, it would be good if the essence of this method is explained briefly.*

**Response:** Thank you very much for your suggestion. Although a full explanation of the B-R model is very long and technical and well-covered in other papers, we have provided a brief summary of the main technique through the inclusion of a high-level algorithm in the next version of the manuscript.[1]
* * *
[1] Line 241: "An example of an asymptotically independent model is the inverted max-stable process (Wadsworth and Tawn, 2012). A general description of all continuous inverted max-stable processes that have standard exponential margins on a spatial domain $X$ is

$$\widetilde{\Omega}(x) = \min_{k \geq 1} U_k / W_k, \quad x \in X, \tag{2}$$

where $U_k$ are points of a unit Poisson process on $(0, \infty)$ and the $W_k(x)$ are independent replicas of a continuous, non-negative stochastic process $W(x)$ in the spatial domain $X$, with $E\{W(x)\} = 1$ for all $x \in X$.

It is convenient to work with a simple inverted max-stable process with unit Fréchet margins, because the marginal distribution can easily be transformed back to the GPD scale. To transform the process $\widetilde{\Omega}(x)$ to unit Fréchet margins, the following transformation is used:

$$\Omega(x) = -\frac{1}{\log\{1 - e^{-\widetilde{\Omega}(x)}\}}, \quad x \in X, \tag{3}$$

then $\Omega(x)$ is an asymptotically independent process with unit Fréchet margins."

**Response to the Reviewer #2**

*The manuscript describes a statistical framework based on an inverted max-stable process allowing to account for the spatial dependence of rainfall across durations. Application is made for a case study in New South Wales, Australia. Using the proposed framework, the author are able to compute conditional and joint return levels of rainfall. Through the use of rainfall ARFs and of an hydrological model, that authors also derive conditional and joint return levels of river flows. Finally the authors derive the failure probability of a highway section, defined as the probability that flood magnitude at any of the five river crossings exceeds a given threshold, assuming a 1-1 correspondence between flood magnitude and rainfall over a catchment.*

*Main comments: The article is well written and mainly clear. The two risk applications of Section 5.1 and 5.2 are very interesting, particularly 5.2 (failure probability of a highway section) which seems to me to be more related to "real" issues than 5.1. The subject is absolutely worth publishing in HESS. However I raise below a couple of major issues to be addressed before publication:*

**Response:** Thank you for your comments. We respond in detail below (your comments in italic font and our responses in normal font).

**Major comment #1:**

*The use of "Intensity-Duration-Frequency curves" in the title seems at the moment misleading. I would have expected from this expression to see e.g. joint or conditional IDF curves at a given station/catchment, i.e. the IF curves for several durations. Here actually only one duration is used for every catchment – basically the concentration time of the catchment. So I'd be tempted to replaced "IDF" in the title (and the text) by "return levels".*

**Response**: As the reviewer comments, the use of "Intensity-Duration-Frequency curves" suggests plots of IF with respect to duration, which we have not shown, and we instead showed return level maps. We propose to use "Intensity-Duration-Frequency relationships" in the title, since the method involves these three elements, but hopefully avoids the suggestion of traditional IDF curves.

The model can produce IDF curves at any given location as well as exceedance relationships of a conditional distribution. We have provided here an additional figure showing this relationship across multiple durations based on the example in Figure 10 of the existing manuscript which focused only on the 9-hour to 36 hour conditional relationship.

[Figure]

**Figure R1.** The exceedance relationship of a conditional distribution across multiple durations based on the example in Figure 8 in the manuscript. The blue line is the relationship between 10-year unconditional return levels (at the location of the blue star in Figure 8) and durations, and the red line is the relationship between one in 10 chance conditional return levels (at the location of the blue star in Figure 8) and durations, given a 20-year event for 36 hr extremes happens at location of the red star (in Figure 8) for the centroid of the Kalang River catchment.

**Major comment #2:**

*I'm puzzled about the GPD fits. If I understood correctly, GPD are fitted to 9 and 36 hr rainfall exceedances. If moving windows are considered, then there is a very strong auto-correlation for both the 9 and 36 hr rainfall values. Have you taken this into account in the fits? A declustering method should be applied. This may be the reason why the fits for 36 hr extremes are usually poorer than for 9 hr extremes (see Figs S5 and S6).*

**Response**: Thank you. We did not consider moving windows; instead, we used restricted time periods for 36 hr rainfall (e.g. 01/01 00:00 to 02/01 12:00; 02/01 12:00 to 04/01 00:00; …). The use of a restricted estimates avoids the need for declustering to undo the effect of a moving window. We used a conversion factor of 1.13 to account for the difference between sliding (unrestricted) $d$ hr rainfall maxima and restricted $d$ hr maxima. This value is based on guidance from Australian Rainfall and Runoff (where Table 2.3.4. from Green et al. (2016) gives the 24-hr factor as 1.15 and the 48-hr factor as 1.11).

Inside the 36 hr period we also restricted the period for 9 hr rainfall (e.g. 01/01 00:00 to 01/01 09:00; 01/01 09:00 to 01/01 18:00; …). This is to align concurrent occurrences of 36 hr and 9 hr rainfall when analysing the spatial dependence across durations. We also used a conversion factor of 1.13 for this period (Figure 5 from Jakob et al., (2005) suggests the fitted conversion factor is relatively stable).

Regarding the fits to the 36 hours extremes, the shape parameter of the GEV has greater uncertainty for some sites (e.g.Fig S5, site 3, 36 hours) which can be seen in the deviations of the observed points from gumbel quantiles. Explanation for variability is unclear to us, but we do not consider it is related to temporal dependence in the extremes.

References used for this response:

Green J, Johnson F, Beesley C, The C, 2016, Chapter 3. Design Rainfall, Book 2 in Australian Rainfall and Runoff - A Guide to Flood Estimation, Commonwealth of Australia

Jakob D., Taylor B.F. and Xuereb K.C. (2005). A Pilot Study to Explore Methods for Deriving Design Rainfalls for Australia - Part 1., HRS No. 10, Hydrometeorological Advisory Service, Bureau of Meteorology, June 2005, (59 pp). http://www.bom.gov.au/water/designRainfalls/hrs10.shtml

**Major comment #3:**

*The part regarding the ARFs seems obscure to me (Section 4.5). Basically I isn't clear tome what the ARF allow for. I interpret between the lines that they allow to transform point return levels to spatial return levels over a catchment. However the way ARFs are described is very confusing to me. For example l. 346 states that "the rainfall extremal estimates need to be converted to the average spatial rainfall using an ARF". First I don't understand what are the "rainfall estimates" (rainfall return levels?). Second I guess that " average spatial rainfall" should be "spatial rainfall return levels". I recommend clarifying Section 4.5 and part of the Introduction dealing with ARFs.*

**Response**: Areal reduction factors (ARFs) were employed to make the adjustment of rainfall depth at a point for a given return level estimate, to an effective (mean) depth over a catchment with the same probability of exceedance as that of the point extreme (Le et al., 2018).

We have clarified the text relating to the explanation of ARFs based on your observations.[2]

References used for this response:

Le, P. D., Davison, A. C., Engelke, S., Leonard, M., and Westra, S.: Dependence properties of spatial rainfall extremes and areal reduction factors, Journal of Hydrology, 565, 711-719, https://doi.org/10.1016/j.jhydrol.2018.08.061, 2018.

**Major comment #4:**

*Expressions such as "10-year conditional return level map given a 20-year event happen at x" are confusing to me. Wouldn't it be less confusing to say this is the levels. expected to occur on average once every 3650 times when a 20-year event happen at x. The "10-year" is misleading to me in that case due to the conditioning.*

**Response**:

On review, we agree that this terminology of return periods is misleading. Our general design intent is introduced as: "What flood flow needs to be used to design a bridge that will fail only once on average every M times that a neighbouring catchment is flooded?" However, we then suggested that if M=10 this
* * *
[2] Line 332: "Before transforming extreme rainfall to flood flow through an event-based model, areal reduction factors (ARFs) were employed to make the adjustment of rainfall depth at a point (i.e. the centroid of a catchment) for a given return level estimate, to an effective (mean) depth over a catchment with the same probability of exceedance as the single point (Ball et al., 2016; Le et al., 2018a)."

implies a 10-year event. On review, we see the use of return periods is confused and are grateful the reviewer has raised the matter.

For the example of daily events (365 days per year), a 10% exceedance of a conditional distribution cannot be used to imply there were 10 years equivalent or 3650 instances – because the condition only applies to a subset of days. As the reviewer has indicated, a descriptive frequency is more transparent and we will remove all instances referring to conditional "return periods". We have exclusively retained descriptive phrases such as "once on average every M times" or "one in M chance" in discussion, figure labels and figure captions.

**Major comment #5:**

*I'm confused with the reference to "annual maxima", whereas the article considers peaks-over-threshold. For example Fig 1 illustrates the case of annual maxima (GEV), which is not the case here. L. 421-423 talks about annual maxima instead of exceedances.*

**Response**: Thank you for pointing this out. We use the peaks-over-threshold model in this paper. So we have fixed the text in L. 421-423, they should be exceedances. We used Fig 1 to shows the limitation of the conventional method so the fact that Fig 1 illustrates the case of annual maxima (GEV) is correct.

**Major comment #6:**

*I haven't understood what is the AEP of Fig 12 and 13. I guess it would be clearer to replace AEP by return periods.*

**Response**: The reviewer is correct that it is not clear what an AEP means for a conditional distribution (as with Major comment #4 for return periods). For example, a 10% chance of exceedance in a conditional distribution is not a 10% *annual* exceedance. For this reason, Fig. 12 is confusing and we have removed it along with associated discussion. The use of AEP in Fig. 13 is correct and we still retain it.

**Minor comment #1:**

*l. 111: Le et al $\rightarrow$ no brackets.*

**Response**: Thank you. We have fixed this.

**Minor comment #2:**

*l. 113 AFR $\rightarrow$ ARF*

**Response**: We have fixed this. Thanks.

**Minor comment #3:**

*l. 116-117: I may be clearer to exemplify (i) in terms of evacuation route design as you do in Section 5.1.*

**Response**: The phrase in question is: "*What flood flow needs to be used to design a bridge that will fail only once on average every M times that a neighbouring catchment is flooded?*"

As with the response to major comment #4, we have addressed the main ambiguity by removing the invalid reference to return periods. Whereas the evacuation route is a general example, phrasing the research question this way allows us to introduce the need for a probability into the design specification.

**Minor comment #4:**

*Fig. 3: add the station numbers 1, 2, 3...*

**Response**: We have fixed this. Thanks.

**Minor comment #5:**

*Fig. 4 estimate conditional rainfall $\rightarrow$ estimate conditional probability rainfall*

**Response**: We have fixed this. Thanks.

**Minor comment #6:**

*l. 277: where $\rightarrow$ to be removed*

**Response**: We have fixed this. Thanks.

**Minor comment #7:**

*l. 294-296: why don't you estimate all parameters (beta, q, c) together?*

**Response**: This method is adopted from the paper of Le et al. (2018). If we fit all parameters ($beta$, $q$, and $c$) jointly, there will be a bias in the estimated $c$ parameter because of the dominance of data points at longer distances, which underestimates the tail dependence coefficients at short distances. The main interest is in short distances, especially at $h = 0$ for the case of dependence between two different durations at the same location (see Figure 8 in the manuscript). Therefore, we estimate beta and $q$ first, and then we use fitted $beta$ and $q$ to estimate $c$.

References used for this response:

Le, P. D., Leonard, M., and Westra, S.: Modeling Spatial Dependence of Rainfall Extremes Across Multiple Durations, Water Resources Research, 54, 2233-2248, doi:10.1002/2017WR022231, 2018.

**Minor comment #8:**

*l. 333-334 it is also noted .. 9 hrs $\rightarrow$ is it useful here?*

**Response**: Yes, it is useful because it indicates that we need to analyse extreme rainfall for different durations.

**Minor comment #9:**

*Section 4.5: to be rewritten to clarify the ARFs as said above*

**Response**: Thank you. We have clarified this.

**Minor comment #10:**

*l. 346: rainfall estimates: what are they?*

**Response**: Thank you. We mean the extreme rainfall intensities at a given location, quantile and duration. We have fixed this in the updated manuscript.

**Minor comment #11:**

*l. 353-354: the BR process → for what duration? With which parameters?*

**Response**: In this paper, we need to calculate areal reduction factors for rainfall of 36 h and 9 h, so we only need to do the simulations for 36 h and 9 h separately. The parameters used are those for the variograms in Eq. (3) for rainfall of each durations, which is $\gamma(h) = \|h\|^\beta / q$ for $q > 0$ and $\beta \in (0,2)$. So we need to fit Eq. (3) separately to observed rainfall of 36 hr and 9 hr to get the fitted parameters. We have provided the explanation for this in the revised version of the manuscript.[3]

**Minor comment #12:**

*l. 360: empirical distributions → I'm confused here. If you use empirical distributions below the threshold, how can you have rainfall at ungauged sites (maps)?*

**Response**: Thank you for your comment. The empirical distributions at ungauged sites are derived through the following steps:

-   Step 1: We use a response surface for threshold for the case study catchments based on covariates including longitude and latitude.
-   Step 2: We use the data of the nearest gauged sites and extract the empirical distributions of rainfall below the interpolated threshold in Step 1.

This method is not perfect, but we think that this is acceptable for this study, and for studies of extremes in general because the non-extremes contribute insignificantly (Thibaud et al., 2013). We have improved the explanation in the revised version of the manuscript.[4]

References used for this response:
* * *
[3] Line 341: "The simulation procedure for spatial rainfall for a given duration is implemented in two steps. In the first step, the theoretical residual tail dependence coefficient function in Eq. (5) is fitted to observed rainfall for the duration of interest to obtain the variogram parameters $q > 0$ and $\beta \in (0,2)$."

[4] Line 349: "The empirical distributions at ungauged sites are derived from the nearest gauged sites using a response surface (latitude and longitude covariates) to spatially interpolate the threshold."

Thibaud, E., Mutzner, R., and Davison, A. C.: Threshold modeling of extreme spatial rainfall, Water 737 Resources Research, 49, 4633-4644, 2013.

**Minor comment #13:**

*l. 373: multiple durations → Is the algorithm of Dombry still applicable in this case? I'm not sure to see how it works for multiple durations.*

**Response**: Yes, we think the algorithm of Dombry works properly for multiple durations in the following way. The covariance matrix of the simulation procedure provided by Dombry is calculated from the variogram in Eq. (4) of our paper. The covariance element for a pair of locations with the same duration (e.g. 36 and 36 hr) is calculated from the variogram of identical durations for 36 and 36 hr. The covariance element for a pair of locations with different durations (e.g. 36 and 9 hr) is calculated from the variogram across durations for 36 and 9 hr.

References used for this response:

Dombry, C., Engelke, S., and Oesting, M.: Exact simulation of max-stable processes, Biometrika, 103, 303-317, 2016.

**Minor comment #14:**

*l. 373 in this case... pair of locations → I don't understand it at all. What covariance matrix are you talking about?*

**Response**: This comment follows from minor comment #13, indicating that we have been ambiguous in this part of the method. We will improve the text to be clearer about how the covariance matrix is constructed.

**Minor comment #15:**

*l. 378 rainfall hyetographs → what rainfall are you talking about? Spatial rainfall over the catchments?*

**Response**: In event-based design methods, template rainfall hyetographs are applied to the areal rainfall total of a catchment for a specified frequency and duration. We have added a brief explanation and reference to design guidelines in the revised version of the manuscript.[5]

**Minor comment #16:**

*Fig. 6: is it useful here? It could be in the supplementary material.*

**Response**: We will move it to the supplementary material.

**Minor comment #17:**
* * *
[5] Line 377: "WBNM calculates flood runoff from rainfall hyetographs that represent the relationship between the rainfall intensity and time (Chow et al., 1988)."

*l. 385 & 387: hydrological models → hydrological model layouts*

**Response**: We will fix this when revising the manuscript.

**Minor comment #18:**

*l. 398: did you apply declustering before estimating the GPDs?*

**Response**: In short, we used estimates based on restricted totals (rather than a moving window) and did not apply declustering. Please also see our response to your major comment #2.

**Minor comment #19:**

*Fig. 7 and SM: there is a huge difference between the extremes at the different stations, e.g. station 2 vs station 6. Could you comment on this? Also what method did you use to produce the confidence bands?*

**Response**: Yes, there is a difference between the extremes at different stations. We can comment on this in the paper. We appreciate it is possible to improve the spatial model with additional covariates (and/or additional data such as daily rainfall observations), but the fidelity of the spatial model is not the main focus of the paper. We feel that the case study is sufficiently plausible to introduce the idea of conditional and joint relationships in hydrologic design.

We used the CAR package in R (qqPlot function). This function produces the confidence bands based on the SEs of the order statistics of an independent random sample (Fox, 2015).

References used for this response:

Fox, J., 2015. Applied regression analysis and generalized linear models. Sage Publications.

**Minor comment #20:**

*l. 421-423: I'm lost here. Do you fit the BR process to annual maxima or exceedances?*

**Response**: Thank you for pointing this out. We fit the BR process to exceedances. We have addressed this in the updated manuscript.[6]

**Minor comment #21:**

*Caption of Fig 8: Abbreviation TDC is useless*

**Response**: Thanks, we have fixed this.
* * *
[6] Line 419: "This is expected, as the dependence at the same site between exceedances at different durations will be lower than between exceedances at the same duration. This is because exceedances of different durations may arise from different storm events (Zheng et al., 2015)."

**Minor comment #22:**

*Fig. 9: I don't understand how you get the maps. For this you need the marginal distribution of rainfall at every pixel. How do you get this?*

**Response**: We get the response surface for the marginal distribution parameters of rainfall at every pixel using a thin plate spline regression against longitude and latitude. We unintentionally omitted these details in the original version, but have included them in the updated manuscript.[7]

**Minor comment #23:**

*l. 469: average spatial rainfall: I'm confused. How can you transform return levels to averages?*

**Response**: We use areal reduction factors ARFs for this conversion and will clarify the text. ARFs a standard design method used to transform an intensity of extreme rainfall at a point to an average rainfall intensity over a spatial domain with an equivalent probability of exceedance (Ball et al., 2016; Myers, 1980; Omolayo, 1993; Shaw et al., 2011; Siriwardena and Weinmann, 1996).

References used for this response:

Ball, J. et al., 2016. Australian Rainfall and Runoff: A Guide to Flood Estimation. © Commonwealth of Australia (Geoscience Australia).

Myers, V.A., 1980. A methodology for point-to-area rainfall frequency ratios. In: Zehr, R.M. (Ed.), Dept. of Commerce, National Oceanic and Atmospheric Administration, National Weather Service. Silver Spring, Md.

Omolayo, A.S., 1993. On the transposition of areal reduction factors for rainfall frequency estimation. J. Hydrol. 145 (1), 191–205. https://doi.org/10.1016/0022-1694(93) 90227-Z.

Shaw, S.B., Royem, A.A., Riha, S.J., 2011. The relationship between extreme hourly precipitation and surface temperature in different hydroclimatic regions of the United States. J. Hydrometeorol. 12 (2), 319–325. https://doi.org/10.1175/2011jhm1364.1.

Siriwardena, L., Weinmann, P., 1996. Derivation of areal reduction factors for design rainfalls in Victoria for Rainfall Durations 18–120 hours. Report, 96(4): 60.

**Minor comment #24:**

*Fig. 11 at the river crossing: which crossing are you talking about? There are several.*

**Response**: Thanks, we have clarified it in the updated manuscript.

**Minor comment #25:**
* * *
[7] Line 438: "In order to obtain the maps in Fig. 7 and Fig. 8, a thin plate spline regression against longitude and latitude was employed to build the response surface for the marginal distribution parameters of rainfall at every pixel."

*l. 495-497: Although Fig 11 … not part of the method → I don't understand these two sentences. What do you mean by "this is not a physical timing difference"?*

**Response**: This text means that our method focuses on the peak of the conditional design hydrograph and does not consider the difference in the timing of the peak. We have improved the explanation to clarify this.[8]

**Minor comment #26:**

*Fig. 12: I don't understand the AEP. Wouldn't it be clearer with return periods instead of AEP?*

**Response**: As with major comment #6, we consider that AEP is a confused term for the conditional probability in Fig. 12. We have removed this figure and associated discussion.

**Minor comment #27:**

*l. 511: extreme rainfall intensity → over a catchment?*

**Response**: Thanks, we have fixed this.

**Minor comment #28:**

*l. 520: and → as a function of?*

**Response**: Thanks, we have fixed this.

**Minor comment #29:**

*Fig. 13: as for Fig. 12, would be clearer to show return periods in the x-axis?*

**Response**: Unlike minor comment #26 focussed on Fig. 12, we think the term "annual exceedance probability" (AEP) is straightforward when applied to the joint probability shown in Fig. 13. The AEP and return period are interchangeable as an inverse relationship, but we expect some readers are more familiar with the terminology of return periods. We have audited our use of these terms throughout the manuscript and will apply a consistent terminology.

**Minor comment #30:**

*Caption of Fig. 13: please explain what are the green segments*
* * *
[8] Line 494: "Although Fig. 9 shows a difference in terms of the time taken to reach the peak flows, the two design hydrographs are separate and this is not a physical timing difference."

**Response**: The green segments are to indicate the interpolation of the individual element failure probability to a system failure probability (discussion line 530). We have added this detail to the figure caption so the description is self contained.[9]

**Minor comment #31:**

*l. 529: 1% annual exceedance prob → 1% AEP*

**Response**: Thank you. We have fixed this.

**Minor comment #32:**

*l. 573: 1.74 → I guess this number depends on the considered levels*

**Response**: Yes, this number depends on the pair of locations that we analyse the conditional probability as well as the considered levels, so we have added a clarification of the considered levels in the revised version of the manuscript.[10]

**Minor comment #33:**

*l. 611: inverted max-stable → inverted max-stable process*

**Response**: Thank you, we will fix it when revising the manuscript.

**Minor comment #34:**

*Fig. S1: I don't understand the figure. Could you please explain what a given point represents? Given Table 1, I would have expected to have points at A=91, 294, 341, 771, 1020, which is not the case.*

**Response**: Fig. S1 provides relationships between areal reduction factors (ARFs) and area (in $km^2$) for different return periods for the case study catchments. These relationships are calculated through the simulation of inverted Brown-Resnick process over equally sized grid points. To get the ARFs for each of subcatchments in the case study (corresponding to area A=*91, 294, 341, 771, 1020*), we need to interpolate these relationships. We will improve the explanation in the revised version of the manuscript.
* * *
[9] Line 527: "The green lines help to interpolate the individual element failure probability from a given system failure probability of 1%. Both horizontal axis and vertical axis are constructed at a double log scale for viewing purposes."
[10] Line 567: "for the two catchments having the strongest dependence (Fig. 7). The corresponding conditional flows were then estimated using a hydrological model WBNM and shown to be strongly related to the ratio of conditional and unconditional rainfall extremes (Fig. 9)."

**Response to the Reviewer #3**

*In general, the paper is well written. However, I have some concerns regarding the real contribution (novelty), connection with the literature and in particular with copula studies, as well as comparison with other models. Main comments:*

**Response:** Thank you for your comments. We respond in detail below (your comments in italic font and our responses in normal font).

**Major comment #1:**

*1. Some important papers related to the topic are missing and more importantly the comparison with them not only in terms of results but also in terms of advantages and drawbacks (e.g. Bardossy and Pegram, 2009, Durocher et al. 2016 and Requena et al. 2018).*

**Response**: Thank you for the suggestion. We have added discussion on these paper to the revised manuscript.[11]

**Major comment #2:**

*2. Regarding the issues motivating the study: the first one seems to be already fixed by Le et al. 2018b (as indicated on page 5), and the second issue is not clear (seems to be written as a statement not as an issue).*

**Response**: Thank you for pointing this out. The second issue relates to the spatial properties of asymptotic dependence (explored in Le et al., 2018a). While these two issues have been separately addressed in previous papers, the contribution is to show how to combine the methods to solve a realistic design problem.

References used for this response:

Le, P. D., Davison, A. C., Engelke, S., Leonard, M., and Westra, S.: Dependence properties of spatial rainfall extremes and areal reduction factors, Journal of Hydrology, Submitted, 2018a.

Le, P. D., Leonard, M., and Westra, S.: Modeling Spatial Dependence of Rainfall Extremes Across Multiple Durations, Water Resources Research, 54, 2233-2248, 2018b.
* * *
[11] Line 59: "Most rainfall models operate at the daily timescale (Bárdossy and Pegram, 2009; Baxevani and Lennartsson, 2015; Bennett et al., 2016b; Hegnauer et al., 2014; Kleiber et al., 2012; Rasmussen, 2013), whereas many catchments respond at subdaily timescales."
Line 47: "Several frameworks have been demonstrated based directly on streamflow observations, including functional regression (Requena et al., 2018), multisite copulas (Renard and Lang, 2007), and spatial copulas (Durocher et al., 2016)."

**Major comment #3:**

*3. The topic can also be closely related to regional frequency analysis or estimation at ungauged basins. The authors did not make this connection or show the difference. In the first case (similarity or connection), a huge literature exists and should be considered.*

**Response**: Thanks for your comment. We have discussed differences to regional frequency analysis and methods of estimation in the revised manuscript.[12]
* * *
[12] Line 72: "Regional frequency analysis is one type of method to estimate IDF curves, where the precision of at-site estimates is improved by pooling data from sites in the surrounding region (Hosking and Wallis, 1997). These methods can be combined with spatial interpolation methods to estimate parameters for any ungauged location of interest (Carreau et al., 2013). To determine an effective mean depth of rainfall over a catchment with the same exceedance probability as at a gauge location, the pointwise estimate of extreme rainfall is multiplied by an areal reduction factor (ARF) (Ball et al., 2016). However, such methods do not account for information on the spatial dependence of extreme rainfall—whether for single storm duration, or for the more complex case of different durations across a region (Bernard, 1932; Koutsoyiannis et al., 1998). The lack of dependence prevents these approaches from being applied to estimate conditional or joint flood risk at multiple points in a catchment or across several catchments, as would be required for a civil infrastructure system."

**Major comment #4:**

*4. The paper focused on a case study (a given set of data). However, the effect of some factors on the performance of the model as not discussed and not studied: for instance, and not limited to, the dimensionality (number of sites) and the size of the subgroups.*

**Response**:

Thanks for your comment. This is beyond the scope of the current study.

**Major comment #5:**

*5. An important missing element from the paper is the notion of copulas which is the most important when dealing with dependence. There is a huge literature in both hydrology and statistics (even in spatial dependence). I'm surprised to not see it in the paper.*

**Response**: We have added literature on copulas into the revised manuscript.[13]

**Major comment #6:**

*6. In section 4: why the GPD is used directly without model selection procedure? Why it is the same for all sites? The GPD is usually asymptotically justified which is not enough (and less justified in hydrology because of the sample sizes) and does not depend on the data at hand. It should be considered as a distribution among others (like GEV for block extremes).*

**Response**: Thank you for this comment. We used the GPD because, in contrast to block maxima, it allows us to consider concurrent rainfall extremes and therefore enables the study of dependence. The intention in this paper is not to work through repetitive fitting of different distributions, but to demonstrate a plausible method based on joint rainfall extremes for the design of linear infrastructure. The same distribution is used at each site with variation at each site carried by the parameters. The marginal model adopted is not perfect, but it is plausible, and sufficient for the intent of showing the application of rainfall dependence to design.

**Major comment #7:**

*7. Lines 245-248: please provide other alternative models and justify the choice of your model.*

**Response**: Thank you. We have added justification of the choice of the Brown-Resnick model in the revised manuscript. For example, Le et al. (2018a) show it has better performance than the extremal-t model.[14]
* * *
[13] Line 91: "Copulas including the extremal-t copula (Demarta and McNeil, 2005), and the Husler-Reiss copula (Hüsler and Reiss, 1989) have also been used to model rainfall dependence."

[14] Line 253: "From Eq. (2), different models for $W$ give different inverted max-stable processes. There are two popular and easily-simulated classes of model for the inverted max-stable processes: the Brown-Resnick model (Asadi et al., 2015; Huser and Davison, 2013; Kabluchko et al., 2009; Oesting et al., 2017), and extremal-t model (Opitz, 2013). This study uses the Brown-Resnick form of equations from the family of an inverted max-stable process because Le et al. (2018a) showed it has better performance than the extremal-t model."

Le, P. D., Davison, A. C., Engelke, S., Leonard, M., and Westra, S.: Dependence properties of spatial rainfall extremes and areal reduction factors, Journal of Hydrology, Submitted, 2018a.

**Major comment #8:**

*8. The assumption, on page 11 line 215, is it reasonable? Is it verified in your case study?*

**Response**: Thank you very much. The assumption of AEP neutrality in rainfall-runoff design is a standard assumption when using IDF curves. While the assumption is in widespread use, it is not without limitation as this issue was explored in to the following two papers.

Bennett, B., Leonard, M., Deng, Y., & Westra, S. (2018). An empirical investigation into the effect of antecedent precipitation on flood volume. Journal of Hydrology, 567, 435-445.

Rahman, A., Weinmann, P. E., Hoang, T. M. T., & Laurenson, E. M. (2002). Monte Carlo simulation of flood frequency curves from rainfall. Journal of Hydrology, 256(3-4), 196-210.

**Major comment #9:**

*9. How the hydrological model (ex. WBNM) is integrated in the steps of fig 4?*

**Response**: The hydrological model (i.e. WBNM) is used to transform the conditional rainfall to conditional flow. A label has been added in the revised version of the manuscript to show this (on the arrow between the see the squares for Section 4.5 and Section 4.6 in the top-right of Figure 4).

**Minor comment #1:**

*1. Fig 4: Why in the independent model, no fitting is required? What it means?*

**Response**: Thank you for pointing this out. The term "the independent model" here is not clear. We have changed it to "the case of independence" and have clarified that we mean the case where rainfall extremes occur independently in space.

**Minor comment #2:**

*2. Sentence from lines 237-240 is long and not clear. Please consider reformulating.*

**Response**: Thank you. We have reworded these sentences in the revised manuscript.[15]
* * *
[15] Line 232: "Without loss of generality it can be assumed that the margins of $Z$ have a unit Fréchet distribution. An important property of dependence in the extremes is whether or not two variables are likely/unlikely to co-occur as the extremes become rarer, as this can significantly influence the estimate of frequency for flood events of large magnitude."

**Minor comment #3:**

*3. Page 13: this text requires to be more accurate about the terms and notation.*

**Response**: Thank you very much. We have clarified this text in the revised manuscript.

**Minor comment #4:**

*4. Lines 287-290: is this case not covered by equation 4?*

**Response**: Thank you. We will rewrite this comment on equation 4. We have clarified that the equation can be used for both cases.

**Minor comment #5:**

*5. All text in page 16 and part of page 17 seems trivial and does not worth all this space. Other more important information deserve this space.*

**Response**: We have removed this material, which will create significantly more space.

**Minor comment #6:**

*6. It is not clear in section 4.6 if the authors consider one hydrological model (WBNM) or other models (see for instance lines 376 and 384).*

**Response**: Thank you for your comment. There is only one type of model (WBNM), but different configurations for each catchment. We have clarified this in the revised text.[16]

**Minor comment #7:**

*7. Line 408 : how you can say the model has reasonable fit? Based on what? And compared to what?*

**Response**: Thank you. We have more explicitly indicated that the comment on fitting relates to Figure 8 (Figure 6 in the updated version). We have also emphasized that the main feature of the model shown in these figures is the relationship at h=0, for the case of dependence between two different durations at the same location.[17]

**Minor comment #8:**

*8. Line 538 : I'm not sure about this statement. It is not true in many situations.*
* * *
[16] Line 385: "Hydrological models (WBNM) for the case study area were developed and calibrated (WMAWater, 2011)."

[17] Line 411: "Figure 6 indicates that the model has a reasonable fit to the observed data given the small number of dependence parameters. Although the theoretical coefficient (red line) does not perfectly at long distances, the main interest is in short distances, especially at $h = 0$ for the case of dependence between two different durations at the same location."

**Response**: Thank you for your comment. We have restricted our commentary to conventional hydrological design that is based on IDF curves, which is more defensible than the original comment which was too general. By construction IDF curves are focused are point-wise estimators of extremes, thus a given design is focused on independent application of univariate statistics.

[revised manuscript text omitted]

---

## Referee Report (RR1)

Review of « Spatially dependent Intensity-Duration-Frequency curves to support the design of civil infrastructure systems »

GENERAL COMMENTS :
The authors have significantly modified and improved the manuscript, taking into account the reviewers' comments. However I still think that the manuscript needs to be improved on three main points before publication (see below for the details and additional comments):
   – The presentation of the inverted Brown-Resnick model is still unclear (Sections 4.2 to 4.4). I think it will hardly be understandable by the readers of HESS.
   – Section 4.5 and 4.6 should be better motivated beforehand. Personally I understood these sections only when reading Sections 5.2 and 5.3. The few sentences at the beginning of Section 4 and Figure 14 are not clear enough for me to understand what are the needed mathematical ingredients.
   – Several equations lack consistency (see below).

DETAILS :
(The lines refer to the marked version)
   – The title hasn't been changed (unlike written in the response to Major Comment 1). Anyway, even with « relationships », I still find the title confusing with regards to content of the article. Why not « Spatially dependent flood probabilities to support ... » ?
   – L 63-64 « to overcome... used » : repetition with the previous sentence
   – L 138 « the lack of dependence » → the underlying independence assumption
   – L 145 « preserve dependence » → account for
   – L 149-150 : could you eleborate more on the difference between copula and max-stable processes ? Why did you choose to use MSP rather than copulas ?
   – L 166 « spatially dependence IDF curves » → I haven't seen such curves in the manuscript
   – - L 262-273: This is a list of what you'll do in the next sections but we don't understand **why** you'll do that (what are the goals?). Please rewrite.
   – L 269 : « to transform conditional rainfall to conditional flows » → This is confusing. I think you transform quantiles, not the absolute values.
   – L 271-273 « An analysis .. comparison » : Actually I don't see any comparison with the independent model (apart in Fig 10). Please remove it from Fig 4 as well.
   – Figure 4: "probability **of** rainfall" , "conditional **probability of** flows", "assume 1:1 relationships **for the probabilities**", joint flood probability → probability of system failure (give the section number)
   – Sections 4.2 to 4.4 should be partly rewritten and reorganized.
   – L 295: please specify that Z is associated to a given duration
   – L 297-298 "without loss... distribution": I don't think that the reader will understand why one can assume that Z is unit Fréchet distributed. The transformation should be given.
   – L 305 "An example … process": Yes but the Gaussian process is another example of AI model. What is the advantage of the inverted BR model with respect to a Gaussian process?
   – L 308-319 "A general … margins": this is not understandable for the great majority of HESS readers. Anyway there is a lack of consistency because tin the construction (2), margins are assumed to be exponential.
   – Eq (4): Again this lacks consistency: written as such, you assume that Z has uniform margins. What is y in the limit? For importantly, what does eta represent **in practice**? This will stay obscure for most of the readers.
   – L 346-360: this is a very complicated way of saying that the dependence depends not only on the distance but also on the duration. Please make it shorter and clearer. The reference to

the time of concentration is confusing because it was nowhere said that you will consider for the duration the time of concentration of the basin. By the way, do you only consider that duration later?

– L 377-385: this part (at least the joint distribution) should come before in Section 4.1. Does Eq. (7): apply to any z1, z2? I guess it applies only to threshold exceedances.

– Eq. (9): I'm confused here. (9) seems to implicitely use $P(Z1>z1)=1/T1$ with z1 the T1 year return level for Z1. However is that true? I though that Z1 and Z2 were threshold exceedances, whereas $P(Z1>z1)=1/T1$ applies if Z1 is an annual maximum, doesn't it?

– L 428-429 "the joint probability … marginals": it could also be specified that in case of independence conditional=marginal.

– Eq. (10): is this useful? I don't think you use it anywhere... Anyway, if you specify this probability in the case of independence, you should also give it for the IBR process.

– L 440: A better title might be "Simulation-based estimation of ARFs"

– L 463-465 "the empirical distributions … thresholds": I don't understand how an empirical distribution can be derived using a response surface since by definition it is not parametric! And what about above the threshold?

– L 474-475 "36 and 6 h durations": only? Other durations are shown in Fig 6...

– L 475-476: "ARF are calulated": I would like to have here a clear explanation on how it is calculated because it is not clear to me.

– L 500-508: I'm a bit lost here because you seem to be able to simulate rainfall in space (see Section 4.5) so why don't you directly simulate rainfall and compute the basin accumulation rather than simulating at the centroid of the catchment and then using the ARF to transform it into a spatial accumulation? My concern is that this may introduce a bias.

– L 553 rainfall extremes → rainfall return levels

– Fig 10: I don't understand what "% AEP" means. Isn't it just "%"?

– L 746: Shouldn't "$P(Z2>z)$" be "$P(Z2>F\_Z(u))$"? Idem in Eq. (A.1)

---

## Referee Report (RR2)

Review of « Spatially dependent flood probabilities to support the design of civil infrastructure systems »

GENERAL COMMENTS :
The authors have significantly modified and improved the manuscript, taking into account my comments. In particular the model is now much more clearly presented for HESS readers. My only significant comment regards the Authors' response to my previous Minor comment #14. Indeed using an inverted max-stable process rather than a Gaussian process -the two of them are AI-complicates much the theory, model estimation and simulation. I know that max-stable processes are theoretically founded for AD models (see Schlather 2002) but what about inverted max-stable processes for AI models? I might be wrong but I don't think there is any theory saying that inverted max-stable process are well-founded for AI models. Given that this article will be mainly read by non-statisticians, I do wonder what is gained by using the inverted max-stable process rather than a Gaussian process, which is much easier to handle. I understand that model comparison is not the goal of the paper but could the authors please better justify their choice for the inverted max-stable model? Otherwise it sound like using a sledgehammer to crack a nut.

DETAILS :
(The lines refer to the marked version)
   – L 69-69: "This is likely to be because" → this is likely because
   – L 102: a more applied work on max-stable process for extreme rainfall is:
     Blanchet, J. & Creutin, J.-D. (2017), 'Co-Occurrence of Extreme Daily Rainfall in the French Mediterranean Region', *Water Resources Research* **53**(11), 9330—9349.
   – L 228: "fit to observed rainfall" → above some large threshold, I guess
   – L 130: "on average only once on average"
   – Figure 4: actually, don't you only fit the marginal model (GPD) above the threshold?
   – L 313-314 and 316-317: isn't this a repetition?
   – L 376: "which the dependence model" → syntax issue
   – L 431-437: "the covariance element … 9 hr" → isn't it possible to make one sentence from these two (for two durations D1 and D2 in general)?
   – L 717: "all of events" → all the events

---

## Author Response (AR2)

**Reference Code:** hess-2018-393

**Title:** Spatially dependent flood probabilities to support the design of civil infrastructure systems

**Corresponding Author:** Phuong Dong Le (The University of Adelaide)

**Contributing Authors:** Michael Leonard and Seth Westra

**Response to the Reviewer**

*The authors have significantly modified and improved the manuscript, taking into account the reviewers' comments. However I still think that the manuscript needs to be improved on three main points before publication (see below for the details and additional comments):*

**Response:** Thank you for your comments. We respond in detail below (your comments in italic font and our responses in normal font).

**Major comment #1:**

*The presentation of the inverted Brown-Resnick model is still unclear (Sections 4.2 to 4.4). I think it will hardly be understandable by the readers of HESS.*

**Response**: The focus of this paper is on application to a design problem, therefore we have focused on explaining key aspects of the application. Rather than seek to repeat or elaborate background theory and definitions of the Brown-Resnick model, we have now simplified the presentation and point readers to papers that give the clearest presentation of the Brown-Resnick model. As a result, Sections 4.2 and 4.3 have been merged, with some theoretical background material and equations removed. Section 4.4 has been moved to the Appendix because it includes necessary calculations for the conditional framework, but otherwise interrupts presentation of the overall framework and application.

**Major comment #2:**

*Section 4.5 and 4.6 should be better motivated beforehand. Personally I understood these sections only when reading Sections 5.2 and 5.3. The few sentences at the beginning of Section 4 and Figure 14 are not clear enough for me to understand what are the needed mathematical ingredients.*

**Response**: Given ambiguity in explanation of the method, we have made a substantially different version of Figure 4. The new flow chart gives a clearer presentation of the key stages and how they are interlinked. The items in the flowchart now mirror the presentation of material in Sections 5.2 and 5.3 to improve consistency of presentation in Section 4. Excerpt text from Line 197-209:

> This section describes the method used to estimate the conditional and joint probabilities of streamflow for civil infrastructure systems based on rainfall extremes, with the sequence of steps illustrated in Fig. 4. The overall aim is to estimate rainfall exceedance probabilities and corresponding flow estimates that account for dependence across multiple catchments. The generalized Pareto distribution (GPD) is used as the marginal distribution to fit to observed rainfall for all durations at each location (Section 4.1). An extremal dependence model is required to evaluate conditional and joint probabilities. Here, an inverted max-stable process is used with dependence not only in space but also in duration (Section 4.2). The fitted model is evaluated in a range of contexts, including the construction of joint and conditional return level maps. The derivation of areal reduction factors and joint rainfall estimates are made with the assistance of simulations based on the fitted model (Section 4.3). An event-based rainfall-runoff model is employed in Section 4.4 to transform extremal design rainfalls to corresponding flows.

[Figure]

**Figure 4.** The flow chart for the overall methodology.

**Major comment #3:**

*Several equations lack consistency (see below).*

**Response**: Specific comments have been given to each point raised below. Some of the background theory has been removed for simplification. Some of the equations have been moved to an appendix to avoid interrupting overall methodology. The unit Frechet transformation has been added, the tail dependence equation has been updated.

**Minor comment #1:**

*– The title hasn't been changed (unlike written in the response to Major Comment 1). Anyway, even with « relationships », I still find the title confusing with regards to content of the article. Why not « Spatially dependent flood probabilities to support ... » ?*

**Response**: We have changed the title to "Spatially dependent flood probabilities to support the design of civil infrastructure systems"

**Minor comment #2:**

*– L 63-64 « to overcome... used » : repetition with the previous sentence*

**Response**: We have removed this sentence.

**Minor comment #3:**

*– L 138 « the lack of dependence » → the underlying independence assumption*

**Response**: We have changed this.[1]

**Minor comment #4:**

*– L 145 « preserve dependence » → account for*

**Response**: We have changed this.[2]

**Minor comment #5:**

*– L 149-150 : could you eleborate more on the difference between copula and max-stable processes ? Why did you choose to use MSP rather than copulas ?*

**Response**: We identify that it is equally possible to use copulas such as the Gaussian copula parametrised as a function of distance.[3]
* * *
[1] Line 79: The underlying independence assumption prevents these approaches from being applied to estimate conditional or joint flood risk at multiple points in a catchment or across several catchments, as would be required for a civil infrastructure system.

[2] Line 85 . This is particularly challenging given that it is not only necessary to account for dependence of rainfall across space, but also to account for dependence across storm burst durations, as different parts of the system may be vulnerable to different critical duration storm events.

[3] Line 236 This study uses an asymptotically independent model, of which there are multiple types including the Gaussian copula (Davison et al., 2012) and inverted max-stable processes (Wadsworth and Tawn, 2012).

**Minor comment #6:**

– *L 166 « spatially dependence IDF curves » → I haven't seen such curves in the manuscript*

**Response**: The manuscript presents IDF maps, but as the reviewer notes, did not present IDF curves. We have updated the title to indicate 'flood probabilities'. Within the paper we have used the phrase *"IDF estimates"* and avoid reference to IFD curves since they are not explicitly presented.

**Minor comment #7:**

– *L 262-273: This is a list of what you'll do in the next sections but we don't understand why you'll do that (what are the goals?). Please rewrite.*

**Response**: We have significantly modified Figure 4 as well as restructured Section 4 to provide greater clarity on the stages of the method and have indicated the goal of the approach.[4]

**Minor comment #8:**

– *L 269 : « to transform conditional rainfall to conditional flows » → This is confusing. I think you transform quantiles, not the absolute values.*

**Response**: The rainfall quantile from the conditional map is used for the magnitude of a design storm. The storm has an associated temporal pattern determined by the national guidelines for hydrological design (Australian Rainfall and Runoff). The absolute rainfall values of this storm are transformed by the model into a flow hydrograph.[5]

**Minor comment #9:**

– *L 271-273 « An analysis .. comparison » : Actually I don't see any comparison with the independent model (apart in Fig 10). Please remove it from Fig 4 as well.*

**Response**: Figure 4 has been updated with mention of the comparison removed.

**Minor comment #10:**

– *Figure 4: "probability of rainfall" , "conditional probability of flows", "assume 1:1 relationships for the probabilities", joint flood probability → probability of system failure (give the section number)*

**Response**: We have updated Figure 4.
* * *
[4] Line 199: The overall aim is to estimate rainfall exceedance probabilities and corresponding flow estimates that account for dependence across multiple catchments.

[5] Line 341: The rainfall extremes are estimated at the centroid of the catchment, and are converted to average spatial rainfall using the simulated ARFs described in Section 4.3. Design rainfall hyetographs are used to convert the rainfall magnitude to absolute values through the duration of a storm following standard design guidance in Australia (Ball et al., 2016).

**Minor comment #11:**

– *Sections 4.2 to 4.4 should be partly rewritten and reorganized.*

**Response**: We have rewritten Section 4. Figure 4 has been updated and Section 4 has been rewritten for greater consistency with the new figure. Section 4.2 has been trimmed to remove background theory of the BR model in preference for references. Section 4.2 and 4.3 have been merged and made more concise to focus on fitting the dependence model. Section 4.4 has been moved to an appendix given the detailed nature of the equations so that the method can focus more on the structure of the model.

**Minor comment #12:**

– *L 295: please specify that Z is associated to a given duration*

**Response**: The text has been updated to indicate it is for a given duration.[6]

**Minor comment #13:**

– *L 297-298 "without loss... distribution": I don't think that the reader will understand why one can assume that Z is unit Fréchet distributed. The transformation should be given.*

**Response**: The transformation equation is now provided in Appendix B.[7]

**Minor comment #14:**

– *L 305 "An example … process": Yes but the Gaussian process is another example of AI model. What is the advantage of the inverted BR model with respect to a Gaussian process?*

**Response**: It is possible to use the Gaussian copula as an asymptotic independent model. Beyond the fit of the dependence model to the data, there is no significant advantage in using one over the other. The focus of this paper is on the ability to construct conditional IDF maps and subsequent design flow estimates rather than a comparative evaluation of models.
* * *
[6] Line 243: For a generic continuous process $Z_i$ for a given duration and associated with a specific location $x_i$, the empirical pairwise residual tail dependence coefficient $\eta$ for each pair of locations $(x_1, x_2)$ is …

[7] Line 539: The unit Fréchet transformation is given as

$$
z = \begin{cases}
\left( log\left\{ 1 - \Phi_u \left( 1 + \frac{\xi(y-u)}{\sigma_u} \right)^{-1/\xi} \right\} \right)^{-1} & y > u, \xi \neq 0 \\
-\left( log\left\{ 1 - \Phi_u exp\left( -\frac{y-u}{\sigma_u} \right)^{-1/\xi} \right\} \right)^{-1} & y > u, \xi = 0 \\
-\{ log F(y_i) \}^{-1} & y \leq u
\end{cases}
\tag{B.1}
$$

where $y$ is the original marginal value and $z$ is the Fréchet transformed value and all other parameters correspond to the GPD specified in Section 4.1. For values below the threshold, $F$ is the empirical distribution function of $y$, $F(y_i) = i/(n+1)$ where $i$ is the rank of $y_i$ and $n$ is the total number of data points.

**Minor comment #15:**

– *L 308-319 "A general … margins": this is not understandable for the great majority of HESS readers. Anyway there is a lack of consistency because tin the construction (2), margins are assumed to be exponential.*

**Response**: The background theory has been removed and is treated in detail within the cited literature. The paper focuses more on the scope of model application.

**Minor comment #16:**

– *Eq (4): Again this lacks consistency: written as such, you assume that Z has uniform margins. What is y in the limit? For importantly, what does eta represent in practice? This will stay obscure for most of the readers.*

**Response**: Thank you. We have fixed the Eq (4), $y$ in the limit should be $z$.[8]

$\eta$ is the residual dependence coefficient, which is a bivariate concept and is defined to measure residual dependence between two asymptotically independent random variables.

**Minor comment #17:**

– *L 346-360: this is a very complicated way of saying that the dependence depends not only on the distance but also on the duration. Please make it shorter and clearer. The reference to the time of concentration is confusing because it was nowhere said that you will consider for the duration the time of concentration of the basin. By the way, do you only consider that duration later?*

**Response**: The text in this section has been made more concise to omit reference to the time of concentration and say that the dependence depends on the duration.[9] The method is for any given duration, the case study application identifies relevant durations based on the time of concentration of the basins. [10]

**Minor comment #18:**

– *L 377-385: this part (at least the joint distribution) should come before in Section 4.1. Does Eq. (7): apply to any z1, z2? I guess it applies only to threshold exceedances.*

**Response**: Equations 7 to 10 have been moved to Appendix B so they do not interrupt the focus on fitting the dependence model and subsequent application. Equation B.1 in the Appendix provides the Frechet transform which indicates that Z refers to values both above the threshold and below.
* * *
[8] Line 243: For a generic continuous process $Z_i$ for a given duration and associated with a specific location $x_i$, the empirical pairwise residual tail dependence coefficient $\eta$ for each pair of locations $(x_1, x_2)$ is

$$\eta(x_1, x_2) = \lim_{z \to \infty} \frac{\log P\{Z_2 > z\}}{\log P\{Z_1 > z, Z_2 > z\}}. \tag{2}$$

The value of $\eta \in (0,1]$ indicates the level of extremal dependence between $Z_1$ and $Z_2$ (Coles et al., 1999), with lower values indicating lower dependence. An example of how to calculate the residual tail dependence coefficient is provided in Appendix A for a sample dataset.

[9] Line 259: The inverted max-stable process is fitted to the observations by minimizing the sum of the squared errors of the residual tail dependence coefficients. When the extreme rainfall at location $x_1$ and $x_2$ are of different durations, the dependence is less than when the extremes are of the same duration. For example, at a single location ($h = 0$), when the duration is the same, the rainfall values are identical and have perfect dependence, but when the duration of extremes are different the values are not identical and the dependence is less. An adjustment needs to be made to the theoretical pairwise residual tail dependence coefficient function when extreme rainfalls have different durations.

[10] Line 192: ... this study assumes a time of concentration of 9 hr for the Deep Creek catchment, while identical times of concentration of 36 hr are assumed for the other four catchments.

**Minor comment #19:**

*– Eq. (9): I'm confused here. (9) seems to implicitly use P(Z1>z1)=1/T1 with z1 the T1 year return level for Z1. However is that true? I though that Z1 and Z2 were threshold exceedances, whereas P(Z1>z1)=1/T1 applies if Z1 is an annual maximum, doesn't it?*

**Response**: Thank you for pointing this out. We have updated the text in the manuscript to indicate that the return periods are calculated on 36 hourly basis. Z1 and Z2 are not restricted to threshold exceedances. The derivation is shown below where the four quadrants of the bivariate space above/below respective thresholds are labelled A, B, C, D. The conditional distribution is the joint divided by the marginal B/(B+C) and where the threshold is z1 is set at *P(Z1>z1)=1/T1*.

[Figure]

**Minor comment #20:**

*– L 428-429 "the joint probability ... marginals": it could also be specified that in case of independence conditional=marginal.*

**Response**: Thank you. We have fixed this.[11]

**Minor comment #21:**

*– Eq. (10): is this useful? I don't think you use it anywhere... Anyway, if you specify this probability in the case of independence, you should also give it for the IBR process.*
* * *
[11] Line 573: For the case that all of events are independent, the joint probability for independent variables is broken down as the product of the marginals, and the conditional probability is equivalent to the marginal probability. When applying Eq. (B.5) for independent variables, the joint probability is therefore calculated by $P(Z_1 > z_1, \dots, Z_N > z_N) = P(Z_1 > z_1) \dots P(Z_N > z_N)$.

**Response**: Eq. (10) is useful when we calculate the joint probability for the case of independence (i.e. the blue line in Fig. 10). We have explicitly referenced it in the method section.[12]

**Minor comment #22:**

– *L 440: A better title might be "Simulation-based estimation of ARFs"*

**Response**: We have restructured Section 4 and title for this section is now 'Simulation based estimation of areal and joint rainfall'

**Minor comment #23:**

– *L 463-465 "the empirical distributions … thresholds": I don't understand how an empirical distribution can be derived using a response surface since by definition it is not parametric! And what about above the threshold?*

**Response**: We use a response surface of threshold for the case study catchments based on covariates including longitude and latitude, i.e. we spatially interpolate the threshold for ungauged sites. For the rainfall above the interpolated threshold, the generalised Pareto distribution in Eq. (1) was used. For rainfall below the interpolated threshold we use the data of the nearest gauged site and extract the empirical distribution.[13]

**Minor comment #24:**

– *L 474-475 "36 and 6 h durations": only? Other durations are shown in Fig 6...*

**Response**: Thank you. In this study, we only need ARFs for 36 and 9h durations due to the time of concentrations of sub-catchments, so we have calculated ARFs for only 36 and 9h durations.[14]

**Minor comment #25:**

– *L 475-476: "ARF are calulated": I would like to have here a clear explanation on how it is calculated because it is not clear to me.*

**Response**: This section has been rewritten.[15] The calculation of ARFs is a substantial step which is covered in detail in Le et al. (2018a). We provide a clearer explanation of the method here in brief, but rely on the reference for detailed explanation of the method. The ARFs are applied here for durations of 36 and 9 hrs.

**Minor comment #26:**

– *L 500-508: I'm a bit lost here because you seem to be able to simulate rainfall in space (see Section 4.5) so why don't you directly simulate rainfall and compute the basin accumulation rather than simulating at the centroid of the catchment and then using the ARF to transform it into a spatial accumulation? My concern is that this may introduce a bias.*

**Response**: The aim of this paper is to develop a method that preserves the traditional IDF framework, where pointwise IDF maps summarise event-magnitudes and separate steps are used to construct rainfall volumes and flow estimates. In
* * *
[12] Line 331: A set of 10,000 years simulated rainfall is generated from the fitted model to calculate the overall failure probability of a highway section (Eq. B.5).

[13] Line 301: For rainfall magnitudes above the threshold the generalised Pareto distribution in Eq. (1) is used, and below the threshold the empirical distribution is used. The empirical distributions at ungauged sites are derived from the nearest gauged sites and using the interpolated response surface of the GPD threshold parameter.

[14] Line 192: ... this study assumes a time of concentration of 9 hr for the Deep Creek catchment, while identical times of concentration of 36 hr are assumed for the other four catchments.

other words, the ARF simulation is a once-off task, whereas application of the overall design method will vary with each context but can utilise the same ARF results.

**Minor comment #27:**

*– L 553 rainfall extremes → rainfall return levels*

**Response**: Thank you. We have fixed this.

**Minor comment #28:**

*– Fig 10: I don't understand what "% AEP" means. Isn't it just "%"?*

**Response**: AEP (Annual Exceedance Probability) is now defined in the text. As an example, a large flood which may be calculated to have a 1% chance to occur in any one year, is described as 1%AEP.[16]

**Minor comment #29:**

*– L 746: Shouldn't "P(Z2>z)" be "P(Z2>F_Z(u))"? Idem in Eq. (A.1)*

**Response**: Thank you. We have fixed this in the manuscript.[17]
* * *

[revised manuscript text omitted]

---

## Author Response (AR3)

**Reference Code:** hess-2018-393

**Title:** Spatially dependent flood probabilities to support the design of civil infrastructure systems

**Corresponding Author:** Phuong Dong Le (The University of Adelaide)

**Contributing Authors:** Michael Leonard and Seth Westra

**Response to the Reviewer**

*The authors have significantly modified and improved the manuscript, taking into account my comments. In particular the model is now much more clearly presented for HESS readers.*

**Response:** Thank you for your comments. We respond in detail below (your comments in italic font and our responses in normal font).

**Major comment #1:**

*My only significant comment regards the Authors' response to my previous Minor comment #14. Indeed using an inverted max-stable process rather than a Gaussian process -the two of them are AIcomplicates much the theory, model estimation and simulation. I know that max-stable processes are theoretically founded for AD models (see Schlather 2002) but what about inverted max-stable processes for AI models? I might be wrong but I don't think there is any theory saying that inverted max-stable process are well-founded for AI models. Given that this article will be mainly read by non-statisticians, I do wonder what is gained by using the inverted max-stable process rather than a Gaussian process, which is much easier to handle. I understand that model comparison is not the goal of the paper but could the authors please better justify their choice for the inverted max-stable model? Otherwise it sound like using a sledgehammer to crack a nut.*

**Response**:

This study uses an asymptotically independent model, of which multiple types are valid including the Gaussian copula (Davison et al., 2012) and inverted max-stable processes (Wadsworth and Tawn, 2012). The inverted max-stable model was ultimately selected in this study to provide consistency with our earlier paper (i.e. "Dependence properties of spatial rainfall extremes and areal reduction factors", in Journal of Hydrology), in which it was demonstrated to preserve the spatial properties of extreme rainfall in an Australian context, including the property of asymptotic independence. Thibaud et al (2013) compared the inverted max-stable model with a Gaussian copula in a case study in Switzerland, and identified that the inverted max-stable model was appropriate.

We note in the manuscript that both models are plausible for asymptotically independent extremes, but in the context of the contribution of this paper (which is an application of joint extremes in an engineering design context), we feel that the inverted-max stable model is well-supported by the data and suitable to illustrate the main concepts.[1]

**Minor comment #1:**

– L 69-69: "This is likely to be because" → this is likely because

**Response**: We have fixed this.
* * *
[1] Line 239: "This study uses an asymptotically independent model, of which multiple types are valid including the Gaussian copula (Davison et al., 2012) and inverted max-stable processes (Wadsworth and Tawn, 2012). The inverted max-stable model was ultimately selected in this study to provide consistency earlier research (Le et al., 2018a), in which it was demonstrated to preserve the spatial properties of extreme rainfall in an Australian context, including the property of asymptotic independence. Thibaud et al. (2013) also compared the inverted max-stable model with a Gaussian copula in a case study in Switzerland, and identified that the inverted max-stable model was appropriate."

Line 500: "Although this study focused on the inverted max-stable model to simulate the extreme rainfall process, other methods such as the Gaussian copula may also be appropriate and should be considered in future applications."

**Minor comment #2:**

*– L 102: a more applied work on max-stable process for extreme rainfall is: Blanchet, J. & Creutin, J.-D. (2017), 'Co-Occurrence of Extreme Daily Rainfall in the French Mediterranean Region', Water Resources Research 53(11), 9330—9349.*

**Response**: We have included this paper into the literature in the updated manuscript.[2] Thanks!

**Minor comment #3:**

*– L 228: "fit to observed rainfall" → above some large threshold, I guess*

**Response**: We have fixed this.

**Minor comment #4:**

*– L 130: "on average only once on average"*

**Response**: We have changed this. Thank you!

**Minor comment #5:**

*– Figure 4: actually, don't you only fit the marginal model (GPD) above the threshold?*

**Response**: Thanks. We have clarified this.

**Minor comment #6:**

*– L 313-314 and 316-317: isn't this a repetition?*

**Response**: Thanks for pointing it out. We have removed the latter one.

**Minor comment #7:**

*– L 376: "which the dependence model" → syntax issue*

**Response**: We have fixed this sentence. Thanks!

**Minor comment #8:**

*– L 431-437: "the covariance element … 9 hr" → isn't it possible to make one sentence from these two (for two durations D1 and D2 in general)?*

**Response**: We think that the current text is OK because the two sentences put emphasis on how to calculate the covariance element for the same duration and for different durations. So, we have decided to keep these sentences.
* * *
[2] Line 90: "Max-stable process has also been used to represent the co-occurrence of extreme daily rainfall in the French Mediterranean region (Blanchet and Creutin, 2017)"

**Minor comment #9:**

*– L 717: "all of events" → all the events*

**Response**: We have fixed this.

[revised manuscript text omitted]